# A scoping review of the impacts of COVID-19 physical distancing measures on vulnerable population groups

Lili Li[1], Araz Taeihagh ●[1] ✉ & Si Ying Tan ●[2]

Most governments have enacted physical or social distancing measures to control COVID-19 transmission. Yet little is known about the socio-economic trade-offs of these measures, especially for vulnerable populations, who are exposed to increased risks and are susceptible to adverse health outcomes. To examine the impacts of physical distancing measures on the most vulnerable in society, this scoping review screened 39,816 records and synthesised results from 265 studies worldwide documenting the negative impacts of physical distancing on older people, children/students, low-income populations, migrant workers, people in prison, people with disabilities, sex workers, victims of domestic violence, refugees, ethnic minorities, and people from sexual and gender minorities. We show that prolonged loneliness, mental distress, unemployment, income loss, food insecurity, widened inequality and disruption of access to social support and health services were unintended consequences of physical distancing that impacted these vulnerable groups and highlight that physical distancing measures exacerbated the vulnerabilities of different vulnerable populations.

The global COVID-19 pandemic had led to around 586.5 million cases and 6.4 million fatalities cumulatively by 10 August 2022, with the United States (US), Brazil, India, Russia, Mexico, Peru and the United Kingdoms (UK) being some of the countries that have been hardest hit in terms of the death toll[1].

With the number of COVID-19 cases and fatalities worldwide still growing, governments have deployed various policy instruments to bring the pandemic under control and to reduce its impact on socio-economic systems. One widely implemented tool in governments' arsenals that has been used to curb the spread of COVID-19 is the deployment of "physical distancing" (often used interchangeably with the term "social distancing") measures. According to the World Health Organization (WHO), social distancing aims to maintain safe physical distancing through decreased crowding[2]. Social distancing (hereafter physical distancing) measures range from lockdowns and school closures to restrictions on social gatherings in homes and public places (Supplementary Text A1).

While policy measures to combat COVID-19 are implemented by governments with the deliberate intention of breaking the virus's transmission chain and bringing the pandemic under control, there are costs involved, including in relation to unintended consequences. The nature of some of these measures, such as nationwide lockdowns, can be draconian and are likely to have some negative repercussions, especially for vulnerable populations.

Researchers have published several systematic reviews of the effectiveness and impacts of physical distancing measures[3–5]. However, a systematic effort to consolidate knowledge on how certain physical distancing measures targeting general populations affect vulnerable groups is lacking. In addition, there is insufficient understanding of how certain targeted physical distancing measures that are intended to ringfence vulnerable populations are designed and

[1]Policy Systems Group, Lee Kuan Yew School of Public Policy, National University of Singapore, Singapore, Singapore. [2]Alexandra Research Centre for Healthcare in The Virtual Environment (ARCHIVE), Department of Healthcare Redesign, Alexandra Hospital, National University Health System, Singapore, Singapore. ✉e-mail: spparaz@nus.edu.sg

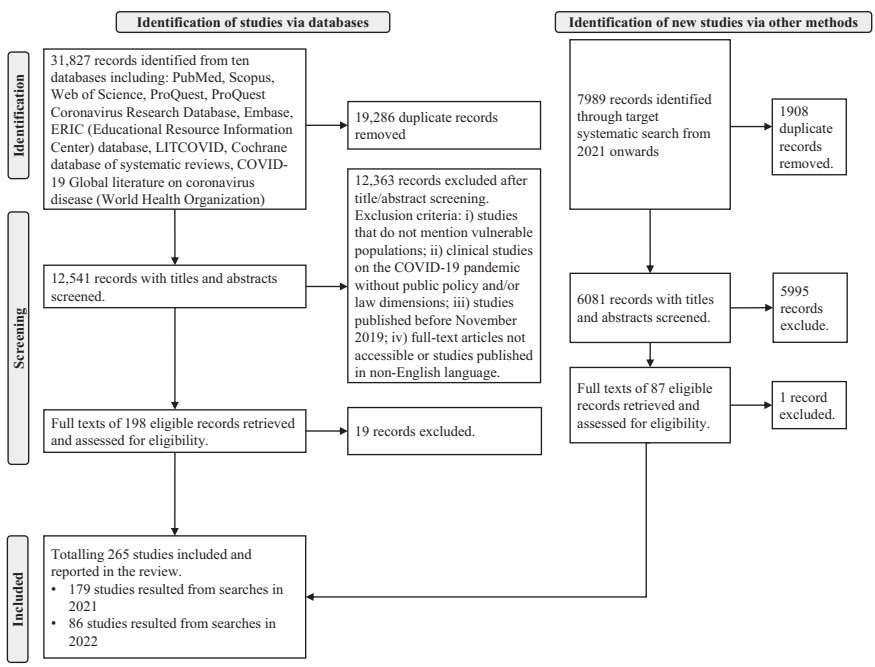

**Fig. 1 | Preferred Reporting Items for Systematic Reviews (PRISMA) flow diagram.** It summarises the data collection and screening processes. We screened 39,816 records and synthesised 265 eligible studies in this review.

implemented in different countries, and how these measures have negatively impacted them. Essentially, ringfencing means protecting something by putting limits on it so that it can only be used for a particular purpose or by a particular group. In the context of this study ringfencing measures resemble "sectoral lockdown so that all the people within a sector or a location can minimise further interactions with the public"[6]. For instance, restricting external visitors to nursing homes and prisons during COVID-19 are some commonly deployed ringfencing measures during the pandemic. Essentially, ringfencing is a form of physical distancing strategy, which aims to limit COVID-19 transmission and contain their spread.

We conducted a scoping review to address the above knowledge gaps, to examine the negative impacts of physical distancing measures on vulnerable populations and identify ringfencing measures designed to counter these impacts.

## Results
### Study contexts and characteristics
Figure 1 summarises the data selection and screening process, according to the PRISMA guidelines[7–9]. Of the total 39,816 records that were produced by searches, we identified 265 eligible studies for synthesis of their results in this review. The list of included studies is available in Supplementary Tables 1, 2, the detailed distribution of studies across countries and vulnerable groups is in Supplementary Tables 3, 4, and additional details of the findings can be found in Supplementary Text A4. Our data cover 49 countries spread across five continents (Fig. 2).

The majority of studies in our dataset focus on children/students and low-income populations (96 studies and 58 studies, respectively). This is followed by studies on the older people ($n = 37$), victims of domestic violence ($n = 16$), people with disabilities ($n = 15$), migrant workers ($n = 14$), refugees ($n = 14$), the people from sexual and gender minorities ($n = 11$), ethnic minorities ($n = 10$), sex workers ($n = 9$), and people in prison ($n = 7$). There are five studies on vulnerable groups in general. The classification we apply is not exclusive, and vulnerabilities can be intertwined, such as children with disabilities or children who are vulnerable to domestic violence.

### Older people: vulnerability and physical distancing
Older people, especially those with chronic diseases, exhibit health vulnerability, including having a high risk of developing severe COVID-19 symptoms if they contract the disease. They are also vulnerable to developing mental health problems, such as anxiety and depression, when isolated at home[10–15]. Some older people are also prone to economic vulnerability, which may exacerbate their health vulnerability due to the resulting decreased access to essential living needs or medical services[16].

In most countries, physical distancing measures have been neutrally applied to all populations, even though they may have negative impacts on the older people. The review included 37 studies on older people. In these studies, physical distancing measures are found to be effective in mitigating COVID-19 infection and deaths among the older people[17,18]. However, at the same time they are found to further predispose older people to greater risks of cardiovascular, autoimmune and neurocognitive diseases[14,19].

The literature also analyses alternatives policy measures to lockdowns, which are more cost-effective. As an example, distributing hygiene kits is considered effective in preventing the spread of COVID-19 while mitigating unintended policy consequences, especially in low- and middle-income countries[20]. To mitigate the risk of COVID-19 outbreaks within nursing homes and long-term care (LTC) facilities, early screening, detection and contact tracing are found to be effective[21]. The distribution of tools for detection and remote monitoring of cases is also found to facilitate COVID-19 control in LTC facilities[22,23].

To address mental health issues faced by older people when socially isolated, the use of technology is considered useful[22–25]. Information and communication technologies are found to help the older people maintain social connections with their loved ones, including attending virtual recreational activities (e.g., playing games) to mitigate loneliness and maintain their wellbeing when they are isolated at home or in LTC facilities[24].

### Children/students: vulnerability and physical distancing
Children are subject to cognitive and communicative vulnerability because of their young age. They are dependent persons (subject to economic vulnerability), relying on their parents/guardians to meet

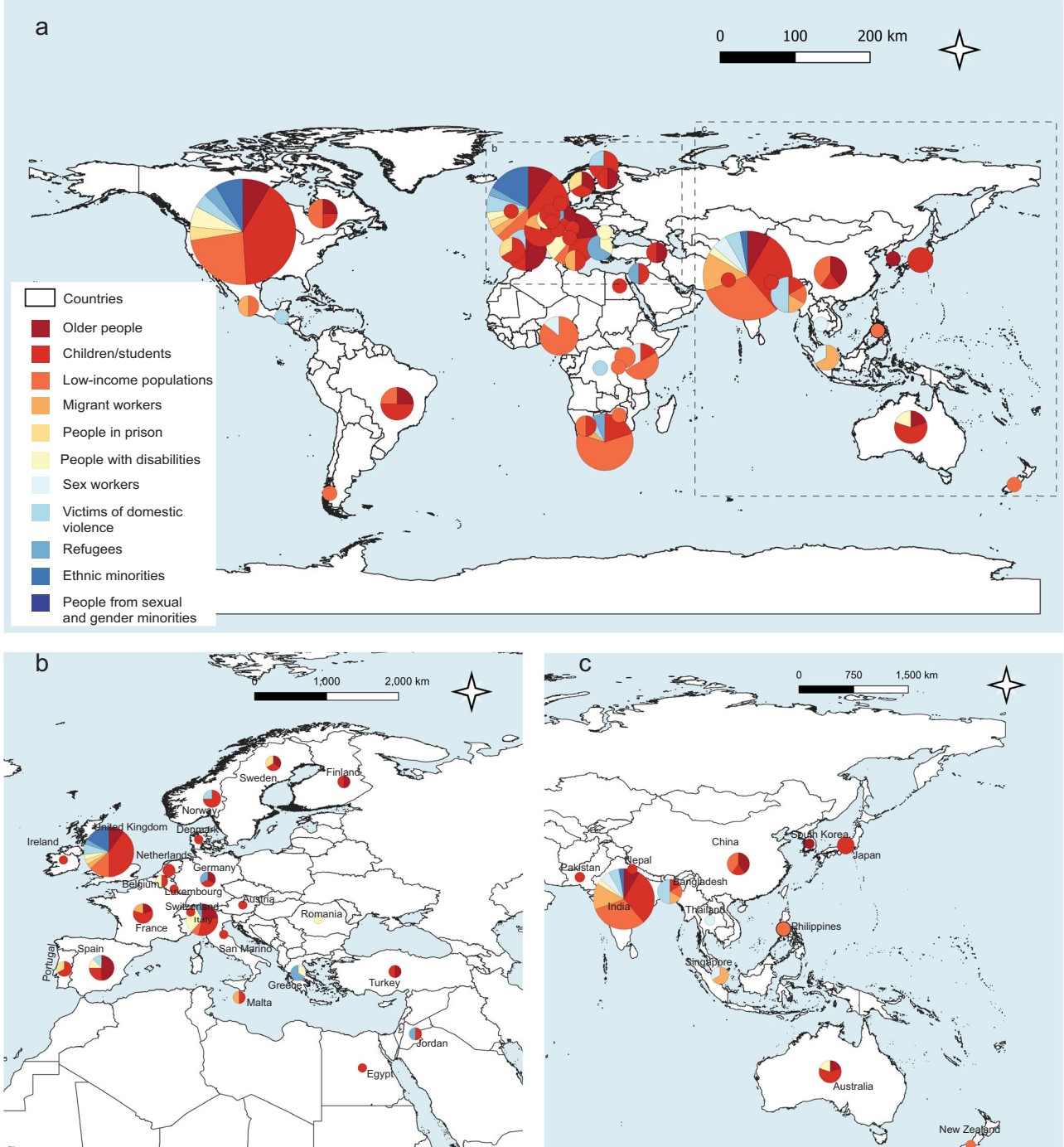

**Fig. 2 | Distribution of included studies across countries and vulnerable groups. a** Global map. **b** Data distribution around Europe. **c** Data distribution around Asia. Data listed in Supplementary Tables 5–7. QGIS version 3.16.12 was used to generate the map (Open Source Geographic Information System, http://www.qgis.org). Global map was downloaded from the OpenStreetMap © OpenStreetMap contributors. Available under the Open Database Licence from: openstreetmap.org.

their living needs and receive financial support[26,27]. Poor economic conditions, such as lack of access to schooling, health and social services, and inadequate clothing, contribute to children's vulnerability[27].

Among the 96 studies on children/students, most studies discuss school closing and remote learning. Unintended detrimental impacts of school closures that are identified in the studies include negative impacts on educational outcomes (differing across learning courses) and on children's physical health (e.g., weight gain)[28,29]. School closures are found to cause widespread concerns about children's suboptimal learning at home[30]. Remote learning is found to exert pressure

on teachers' technical capacities to operate digital teaching, and often increases their working hours[31,32].

Across the world, the implementation of remote learning has sparked concerns about the widening of learning gaps between students from different socio-economic classes. Virtual learning environments cause inevitable disruptions. Children from low-income families are affected more significantly by disruptions in learning and being deprived of the services and support that they receive in schools[33–37]. They are likely to lack access to the internet and other digital resources, such as computers, and face issues such as a lack of

regular meals and parental support, a home environment that is not conducive for learning, and a high risk of exposure to COVID-19 in crowded housing conditions[33,34,36,38,39].

There have been debates on the extent to which school closures are effective in breaking the virus's transmission, and the necessity of reopening schools to facilitate effective learning[40–48]. As indicated by the literature, decisions on school closures or reopening should consider the severity of COVID-19 infections in the community and the ability of schools to enforce physical distancing, as well as to detect and trace suspected cases on campus/in school facilities[43,45]. It has been shown that in many countries school reopening did not necessarily result in increases in COVID-19 transmission if health and safety protocols were adhered to. These protocols include the following: teaching children hygiene measures (e.g., washing hands), mandating mask-wearing, ensuring one- to two-metre distancing, screening at entry, grouping students and reducing class sizes, limiting after-school activities, moving to outdoor classrooms, testing and tracing suspected cases, quarantining confirmed cases, suggesting that parents do not congregate outside the campus gates, staggering drop-off and pick-up times, enhancing air circulation, shortening school hours, offering options for in-person, remote or hybrid learning, and disinfecting facilities[47,49–66].

To address children's mental health issues, studies[67,68] suggested that parents communicate constructively with their children about the current COVID-19 pandemic situation, in accordance with their maturity level and ability to understand the crisis, and in some parts of the world scholars have advocated for schools and families to assess and identify children's mental health needs and issues[69,70].

To enhance remote learning, especially to support vulnerable students from low-income families, solutions that have been applied have ranged from circulating printed materials to be used in learning from home[32], putting out television broadcasts related to courses or other media[71], identifying children's needs and offering target instructions[72–74], and offering support such as helping people obtain laptops, providing helplines, providing free-of-charge data SIM cards and implementing food programmes[75,76]. Many governments have also established funds or fiscal stimulus packages to meet children's educational needs during the COVID-19 pandemic[35,73,74].

## Low-income populations: vulnerability and physical distancing

As the term suggests, low-income populations face adverse financial conditions. They may also face less favourable employment conditions, such as temporary contracts, part-time employment, self-employment and employment in the low-income informal sector. Low-income levels also correlate with poor health[77,78], poor housing conditions and even homelessness[78]. As low-income populations typically have few savings, the loss of their job or income can further worsen their financial situation, causing other health risks, such as food insecurity[79].

Fifty-eight studies across 19 countries find similar results regarding the challenges faced by low-income populations in adhering strictly to various physical distancing measures. These include congregated living spaces, a dearth of sanitation facilities, an inability to work from home (e.g., if they work in the manufacturing and processing industries), a lack of access to job protection or paid leave, food shortages and food insecurity, worsened financial conditions and being forced to live off savings, a lack of regular access to basic hygiene, and a lack of resources and infrastructure for testing, isolation and contract tracing[10,38,78–101]. Lockdowns are reported to be less effective in containing and reducing new COVID-19 cases in low-income countries and communities[97,102–104].

In addition, the social vulnerability of low-income populations intersects with economic vulnerability, and contributes to health vulnerability. For instance, in slums and informal settlements, residents are exposed to poor living conditions, a dearth of sanitation facilities

and a lack of clean drinking water; moreover, the high population density in slums renders one- or two-metre distancing impractical[10,38,86,96,104–106]. These factors have negative repercussions for governments in terms of tracing, testing and treating COVID-19 infections. Those with low incomes also face barriers to accessing healthcare due to income loss and reduced employer-sponsored healthcare[83,101], while their access to government aid or relief packages is limited[84,104,107,108]. The literature also records the mental distress suffered by low-income populations during the COVID-19 pandemic[38,108–110].

According to the literature, to control the spread of COVID-19 among low-income communities, physical distancing measures should be context-specific[90]. Intensive screening, testing, and contract tracing, and the isolation of infected cases, are found to be useful to keep cases low[80,111,112]. Other measures, such as offering affordable masks, the availability of clean water, sanitiser and soaps, and easing crowding in low-income communities, are also necessary[80,105,112].

Greater economic support has been called for to assist low-income populations[35,87,93,96,99,113,114]. It is argued that access to unemployment insurance, paid medical leave and reemployment services should also be strengthened for those on low incomes[79,87,115].

More effective channels for communication between low-income populations and governments can be established[87]. Community-led efforts by non-governmental organisations are found to have been instrumental in aiding low-income populations during the pandemic. Their efforts include offering food, distributing COVID-19 information and education through social media or SMS, facilitating contract tracing and quarantine, delivering groceries and other essential items to patients in quarantine, and establishing longitudinal clinical and social support for them through the involvement of community health workers and public health specialists[106,116–118].

## Migrant workers: vulnerability and physical distancing

A migrant worker is defined as "a person who is to be engaged, is engaged or has been engaged in a remunerated activity in a state of which he or she is not a national"[119]. Migrant workers move away from their usual place of residence across an international border to a different country, or to a different place within the same country[120]. Migrant workers are exposed to economic vulnerability due to experiencing adverse employment conditions, such as precarious recruitment processes, an absence of accurate information on the terms and conditions of employment contracts, and a lack of labour law coverage in the destination state/location[121]. They may also face mental health vulnerability[122], such as suffering from emotional distress due to social and cultural isolation, hardening migration policies and even the loss of accommodation in the destination state/location[123–126].

Owing to lockdowns/stay-at-home orders implemented to control COVID-19, migrant workers (inter-state or inter-country) have faced a variety of challenges, including job losses and income loss[127], food insecurity[127,128], a high risk of exposure to the virus in crowded dormitories[129,130], and limited access to primary healthcare and COVID-19 treatment[107,130]. A study in the UK reported that the pandemic has led to a deterioration in the housing stability of the migrant worker population, making them even more socially and economically isolated, which has created ripple impacts, such as preventing them from accessing remote healthcare services[126]. Migrant workers tend not to qualify for many COVID-19-related social benefits (e.g., food rations) provided by the destination country and are likely to be repatriated to their home countries[131]. Migrant workers also suffer significant mental stress and loneliness due to financial instability, the isolation of migrant worker dormitories, language and cultural barriers, an inability to send remittances to families, and barriers to returning home to see their family members[92,122,129,130]. These predicaments have resulted in some migrant workers adopting different survival

strategies, such as reducing their expenses, consuming less food, and selling their jewellery, land and other possessions[123].

A notable example of the drastic measures that have been taken to address COVID-19 transmission among migrant workers is Singapore. The initial COVID-19 outbreaks in migrant workers' dormitories in the country put Singapore in the global limelight and the government and non-profit organisations responded swiftly by enhancing migrant workers' access to medical support and public health information[129,132]. The government also employed extensive testing, established quarantine facilities, offered financial support for employers to pay migrant workers' salaries, supported communication via facilitating internet connectivity and prepaid phone cards, and provided timely and free-of-charge medical care for infected migrant workers[129]. In addition to governmental efforts, online information about COVID-19 in native languages that were accessible to migrant workers, as well as hotline- and web-based counselling services, were provided by a local non-profit organisation for migrant workers[132].

## People in prison: vulnerability and physical distancing
The primary vulnerability faced by people in prison is institutional or deferential vulnerability, as they are deprived of their liberty. In addition, they are socially isolated and live in overcrowded spaces, which can lead to social vulnerability. Vulnerability to mental health issues[133] and violence[134] are also typical among inmates. The spatial density in prisons has been associated with outbreaks of infectious and communicable diseases[135,136].

Enforcing social isolation or distancing in prisons has been shown to exacerbate inmates' mental health issues and even suicide rates[137]. Poor communication about the COVID-19 pandemic from prison staff contributes to increased stress and anxiety among people in prison[138]. Prison escapes and riots have been reported in Brazil, Italy, Sudan and Nigeria due to increased social isolation measures imposed on people in prison, such as restrictions on receiving visitors[80,137].

Some countries, such as Sweden[139] and the UK[138], have allowed virtual or video visits from families to mitigate the mental distress experienced by people in prison. For instance, in Sweden, the Swedish Prison and Probation Service (SPPS) has provided inmates with tablets to enable phone calls or video calls with their children or their other family members and friends[139]. Many prison units have also organised weekly information meetings to update inmates about the COVID-19 situation and the policy measures taken to preserve their safety in prisons[139]. These measures have been found to be helpful in mitigating inmates' fear of contracting the virus and preserving their mental health.

To reduce population density and reduce the spread of COVID-19 in prisons, some governments have released a number of people from prisons, primarily to home confinement or community supervision, such as in the US, Portugal, Iran, Ireland, Morocco, Libya, Burkina Faso, Uganda and Nigeria[80,137,140,141]. Certain groups of people in prison have been considered for rapid release[137], such as those who are approaching their actual release date, those eligible for medical release, those approved for community supervision, those held on minor charges, and those held pretrial on bail.

## People with disabilities: vulnerability and physical distancing
A disability is defined as any physical or mental condition (impairment) that results in a limitation on a person's activity (difficulty in undertaking certain activities, e.g., difficulty seeing, hearing, walking, or problem-solving) or a restriction on their participation (difficulty participating in normal daily activities, e.g., working, social activities, or accessing healthcare and preventive services)[142]. People with disabilities typically face health vulnerabilities. Economic vulnerability is also a prominent vulnerability reported among the people with disabilities as they often have less access to educational and employment opportunities, and are more likely to live in poverty or on a low income[143,144].

When physical distancing measures are enforced, particularly during lockdowns, people with disabilities face challenges such as reduced social connections and reduced access to healthcare services[145]. For instance, in-person supportive services, such as community-based rehabilitation, may be disrupted[146], and people with mental disabilities (e.g., children with autism) may face worse mental health outcomes due to increased social isolation and suspension of therapy[68,147]. The enforcement of physical distancing measures also inadvertently reduces access to medication, healthcare services and transportation for adults with mobility disabilities[148]. In addition, their reduced mobility and social connections limits their access to social welfare services, such as food rations[109].

In this review, one of the populations with disabilities found to be worst affected is children with disabilities. When schools close during the pandemic, the lack of necessary technologies (e.g., braille readers) and services (e.g., help and support from special education teachers) at home may lead to their experiencing difficulties in participating in remote learning[39]. Studies from Egypt and India have reported that remote learning poses challenges to the caregivers of children who are intellectually challenged, as these children's attention span, impulsivity, mood swings and hyperactivity worsen post-lockdown, thus reducing their performance[149–152]. By and large, it is found to be challenging to provide remote learning to students with disabilities because the services and the number of specialised instruction hours may differ for each student[153].

The studies included in the review discuss solutions to help children with disabilities during the COVID-19 pandemic, such as allowing them to borrow learning devices from their school[39], offering financial support to meet their educational needs[154] and encouraging parents to help their children to understand the COVID-19 pandemic[68].

## Sex workers: vulnerability and physical distancing
By the nature of their occupation, sex workers face health vulnerability, due to their susceptibility to sexually transmitted infections, and social vulnerability, such as increased risk of being subjected to violence[155]. Some also face economic vulnerability due to adverse financial conditions[156].

In the context of the COVID-19 pandemic, sex workers living with HIV are reported to have experienced a lack of access to testing and treatment for sexually transmitted infections during lockdown periods[157–159]. In addition, they also have limited access to COVID-19 social services and safety nets offered by governments[160]. A study in Singapore also revealed that sex workers have experienced greater economic hardship during the pandemic as a result of a reduction in the demand for sex workers. This phenomenon has also caused the out-migration of sex workers and a shift of sex work towards online spaces[161].

To meet the basic needs of sex workers during the COVID-19 pandemic, community-based organisations have stepped up efforts to provide food, financial aid, COVID-19 safety guidance, and community-driven health interventions, including anti-retroviral therapy (ART)[157,162]. In India, considering that ART was disrupted, the organisation Ashodaya Samithi formed a community-led system to distribute ART at private and discreet sites and utilised WhatsApp messaging to share information related to the pandemic[158]. In Thailand, to secure the basic hygiene and personal protection needs of sex workers, community-led organisations, including the Raks Thai Foundation, Dannok Health and Development Community Volunteers, and SWING, provided food, hand sanitisers, condoms and face masks to the sex worker population[157]. In Africa, community-led outreach has distributed food packs to sex workers during the pandemic[163].

**Victims of domestic violence: vulnerability and physical distancing**

When isolated at home, victims of domestic violence are more socially isolated and have reduced access to institutional support[164]. In many cases, having a disability can exacerbate a person's vulnerability to domestic violence[165,166].

Victims of domestic violence may face difficulties in leaving home to access institutional support during lockdowns. In Spain, the Democratic Republic of Congo, South Africa, Guatemala, India, and Bangladesh, lockdown periods saw increased incidence of, and police reports of, domestic violence against women[167–172]. Studies surmise that the official figures for such violence are under-reported as many formal and informal communication channels to access help for the victims, including transport to access shelters in some countries, either shut down or slowed down their operations during the pandemic, fostering change in help-seeking behaviour among abused women[167,169,173]. School closures that force children to be homebound also put them at a higher risk of violence[174]. A study in a city in Wales, in the UK, reported a significant increase in child protection medical examinations through self-referrals and third-party referrals in 2020 as compared to 2019[175].

Remote counselling services through virtual platforms can be helpful for victims of domestic violence[10]. In the UK, due to alarming trends in domestic violence during the pandemic, several national campaigns were organised to raise awareness of domestic abuse and to highlight available help services[176]. Helpline services were extended for victims of domestic violence in the UK, and the government provided funds to support these helplines and other online support services from April 2020[176]. Victims of domestic violence have sometimes been allowed to bypass lockdown restrictions to travel to sheltered accommodation to seek refuge[176]. Schools and universities have also extended accommodation to students who are vulnerable to domestic violence and have ensured student counselling and support services are provided to support them[176]. In France and Italy, governments also commissioned hotels as shelters for victims of domestic violence[164].

**Refugees: vulnerability and physical distancing**

The vulnerabilities faced by refugees, as identified in the literature, include economic vulnerability[177], social vulnerability (e.g., social isolation, living in crowded informal refugee camps, a lack of water and sanitation facilities, and vulnerability to violent attacks)[178,179], and health vulnerability, particularly mental health problems[180].

During the pandemic, precarious and overcrowding housing conditions have made it difficult for refugees to maintain physical distancing or self-isolation at home when required[181]. Owing to the shutting of government services and decreased numbers of volunteers working in refugee camps during the pandemic, refugees have experienced limited access to food, basic sanitation and medical care[181–183]. This situation exacerbates the mental health issues of refugees, who may already be living with post-traumatic stress disorders or other mental illnesses. For some refugees, owing to the sensitive nature of their trauma histories, their forced isolation may bring to the surface traumatic memories of the past[181–183].

Solutions, such as encouraging mask-wearing, limiting mobility, sectoring and setting up quarantine areas, as well as quickly detecting and isolating suspected or confirmed cases, have been implemented to limit the spread of COVID-19 in refugee camps[80,184–186]. Mental health services have been provided for refugees through phone or video conferencing emotional therapies, as well as through other voluntary mental health services in a targeted attempt to reach out to refugee families[182,187,188]. The active deployment of social workers to areas where asylum-seekers reside has also helped to address the social and psychological vulnerabilities they face during the pandemic[189].

**Ethnic minorities: vulnerability and physical distancing**

Ethnic minority groups typically face social vulnerability as they are socially isolated/marginalised. They tend to have less access to education and health services than the ethnic majority[190]. Ethnic minorities also face economic vulnerability. In any given country, members of an ethnic minority are more likely to live in poverty than the ethnic majority, on average, and are less likely to work in high- or semi-skilled jobs[190].

Lockdowns may be particularly challenging for ethnic minority groups as they are less able to work from home or to self-isolate at home[100]. People from an ethnic minority background are more likely to work in adverse employment conditions and to face greater financial concerns than the ethnic majority[93,113]. Physical distancing measures are also found to substantially impact the mental health of ethnic minorities[191]. Language barriers experienced by ethnic minorities are also highlighted as an issue. These barriers can hamper their understanding of the pandemic and can hamper government efforts to enforce physical distancing measures[192].

**People from sexual and gender minorities: vulnerability and physical distancing**

The people from sexual and gender minorities are socially isolated[193] or vulnerable to violence;[194] they also have health vulnerabilities, particularly vulnerability to HIV risks[193,194] and mental health illnesses[195]. The marginalised social identities of sexual and gender minorities reinforce and intersect with their health vulnerability. This review finds that physical distancing measures to control COVID-19 may exacerbate the social and health vulnerabilities of the people from sexual and gender minorities[195–198]. Most importantly, enforcing physical distancing may limit their access to essential medical services, including HIV testing and treatment[196,197]. For those who are vulnerable to mental illnesses related to discrimination and lack of family acceptance, the enforcement of physical distancing measures directly cuts off their access to supportive friends and partners, reduces their sense of social connectedness and aggravates feelings of loneliness[198]. For instance, a study in Brazil revealed that transgender populations reported substantial mental health problems and challenges in accessing healthcare during the pandemic[199]. The stigma and social exclusion of the transgender population were exacerbated during the pandemic, especially those who were older, as their social and health needs were not properly addressed due to the small and dispersed nature of this population[200].

**Summary of results**

Table 1 summarises the findings and specific examples mentioned in the literature (Supplementary Text A4.12).

## Discussion

While there is a robust scientific basis for enforcing physical distancing measures to slow the transmission of COVID-19, little is known about the ethical implications and socio-economic trade-offs associated with such measures. Physical distancing measures may disproportionately and negatively impact the most vulnerable groups in society through job losses, reduction in incomes, a deterioration in mental health and a widened socio-economic gap between the richest and the poorest[201–203].

The review has revealed the negative impacts that physical distancing measures can have on different vulnerable populations (Table 1). The enforcement of physical distancing affects the utilisation of and access to essential health services among the older people, people with disabilities and people from sexual and gender minorities. Mental distress caused by social isolation is found to be common among different vulnerable populations. Effective communication mechanisms should therefore be established to make public information regarding the COVID-19 pandemic and related policy measures

**Table 1 | Summary of findings across vulnerable populations**

| Vulnerable groups | Negative impacts of physical distancing | Ringfence measures | Exemplar countries |
|---|---|---|---|
| Older people | • Decreased utilisation of hospital services.<br>• Worse mental health outcomes. E.g., prolonged loneliness.<br>• Delayed health visits for other non-communicable diseases such as cardiovascular and neurocognitive diseases. | • Use of technology to maintain social connections. E.g., New Brunswick (Canada).<br>• Deployment of volunteers to provide company and check for safety. E.g., New Brunswick (Canada).<br>• Early detection and contact tracing in LTC facilities. E.g., South Korea. | • Brazil and France: discussions held to continue to isolate the older people while lifting the physical distancing measures for others[216,217].<br>• The UK, US and India: isolation at homes was reported to have negatively affected the older people's mental health and resulted in prolonged loneliness due to limited social activities[10,11,19,218].<br>• Europe: greater stringency of physical distancing was associated with worse mental health outcomes among the older people[15].<br>• Germany: there were decreased utilisation of hospital services[219].<br>• A study in India reported that income loss was more significant among the older migrant workers than other age groups during lockdown[127].<br>• South Korea conducted a nationwide surveillance of 1470 LTC facilities in 2020 to monitor compliance with physical distancing rules, including identification and isolation of patients with COVID-19 symptoms and the quarantining of employees who had recently travelled to high-risk countries[21].<br>• New Brunswick, Canada: the government provided one iPad for every 10 residents residing in LTC facilities and deployed volunteers to support them[22].<br>• India: online shopping and digital government/banking services were leveraged to make the older people less dependent on others[24]. |
| Children/students | • Learning disruption.<br>• Decreased social interactions.<br>• Significant learning disparities between children from high vs. low-income families. | • Printed materials provided to students with insufficient technical resources. E.g., Germany, Austria, and Switzerland.<br>• Education funds. E.g., the US, Italy.<br>• Childcare support programme for parents. E.g., South Korea.<br>• Remote learning support offered by NGOs. | India, Italy, Japan, Nepal, Pakistan, Spain, South Africa, Switzerland, the US, the UK, and Turkey had reported that school closures created mental health issues among the children[32,37,69-71,92,220-230].<br>• In India, Pakistan, South Africa and the US, mental health support programmes were established to help children with mental health issues when isolated at home[37,69,224,229].<br>• In Australia and the US, difficulty in adjusting to remote learning was reported[68,153,154]. In addition, students with disabilities, such as the braille readers, may not have essential technologies to facilitate their learning at home[153].<br>• US: variations in the types of services needed and the length of specialised instructions required for children with disabilities rendered online learning challenging[153].<br>• South Korea: childcare programmes developed for parents who needed help with childcare due to school closures[231].<br>• Italy: the "Cura Italia" (Care Italy) Decree-Law (no.18/2020) mandated 85 million euros be allocated to schools to enhance remote learning, with more than half of the funds earmarked for digital devices to low-income families[39].<br>• Australia: individualised funding schemes such as the National Disability Insurance Scheme offered financial support for families to access technologies to facilitate remote learning for children[154].<br>• South Africa: NGOs stepped up to help students with remote learning[131]. |
| Low-income populations | • Job and income loss.<br>• Lack of access to food, clean drinking water and sanitation facilities.<br>• Mental distress | • Case detection, tracing and treatment enhanced.<br>• Access to health services enhanced.<br>• Food and other necessities offered by governments and NGOs. | .• Andhra Pradesh in India: slum residents reported mental distress during lockdown[110].<br>• Dharavi, a large-scale slum in India, flattened the curve quickly through screening, contact tracing, and quarantine measures[106].<br>• India, China, and Nigeria: NGOs worked with governments to offer food and other necessities (e.g., fuel)[85,93,104,106].<br>• India and South Africa: government transferred funds were found to be helpful for the low-income populations[93,114] |
| Migrant workers | • Job and income loss.<br>• Overcrowded living environment where physical distancing is not feasible.<br>• High risks of exposure to COVID-19 infection.<br>• Unable to receive rapid treatment. | • Transport facilities arranged for migrant workers to return to their families. E.g., India.<br>• Effective testing and quarantine measures.<br>• Access to physical and mental health services enhanced. | • UK: housing instability deteriorated for migrant workers during the lockdown[126].<br>• Mexico: Mental health programmes organised by NGOs had to stop due to physical distancing[123].<br>• India: inter-state migrant workers encountered various challenges during the lockdown[92,122,127,128]. The migrant workers faced travel restrictions and |

**Table 1 (continued) | Summary of findings across vulnerable populations**

| Vulnerable groups | Negative impacts of physical distancing | Ringfence measures | Exemplar countries |
|---|---|---|---|
| | • Barriers to return home due to travel restrictions, leading to prolonged social isolation. | | attempted to walk back to their home villages from the cities but were arrested in various inter-state borders for violating the lockdown mandate in March 2020[127]. <br>• Singapore: the government and NGOs such as HealthServe put in place a bundle of policy measures to successfully protect migrant workers after the initial COVID-19 outbreaks in dormitories[129,132]. These include extensive testing, establishing quarantine facilities, offering financial support for employers to pay migrant workers' salaries, providing internet connectivity and prepaid phone cards, and free medical care for infected migrant workers[129]. Hotline and web-based counselling services and a multi-lingual website containing information about COVID-19 were also established[132]. |
| People in prison | • Crowded living environment. <br>• Difficulty in enforcing physical distancing. | • Releasing a proportion of people in prison. <br>• Rapid detection and quarantine of suspected cases. <br>• Tablets provided for inmates to enable phone calls or video calls with their children or other family members and friends. <br>• Information meetings regularly organised to update inmates about the COVID-19 situation and the policy measures taken to preserve their safety in prisons. | • In the US, Portugal, Iran, Ireland, Morocco, Libya, Burkina Faso, Uganda and Nigeria, the sentences of a proportion of people in prison serving time in jail were converted to home confinement or community supervision[80,137,140,141]. <br>• Prison escapes and riots were reported in the US, Brazil, Italy, Sudan and Nigeria due to increased social isolation measures such as restrictions on visitors[80,137,232]. <br>• Portugal: about 10% of the people in prison was released on short notice, including people in prison aged 65 years and above with underlying health conditions[141]. They were released with no means of transport to return home when physical distancing measures such as restrictions on public transport had been already implemented[141]. <br>• Sweden was quick to detect and quarantine suspected cases in prison. The Swedish Prison and Probation Service provided inmates tablets to enable phone calls or video calls with their family members and friends. Weekly information meetings were organised to update inmates about the COVID-19 situation and ensure their safety in prisons[139]. <br>• UK: prisons allowed virtual or video visits from families[138]. |
| People with disabilities | • Decreased social connections. <br>• Lack of access to healthcare, emotional support, and transportation. <br>• Lack of necessary technologies & services at home for students with disabilities. | • Delivery of distance learning and related services for students with disabilities. | • Egypt, and India: remote learning posed a challenge for children who are intellectually challenged (e.g., worsened attention span, and hyperactivity)[149–152]. <br>• US: school districts modified instructions and learning goals to account for the limitations of remote learning, held virtual meetings with school officials, parents and students, and increased collaborations between teachers and parents. However, these measures left out families without computers[153]. |
| Sex workers | • Reduced access to testing and treatment for sexually transmitted infections. <br>• Reduced access to anti-retroviral drugs for HIV-infected sex workers. | • Community-driven health interventions through outreach services to provide personal protection and hygiene equipment. | • India: community-based organisations offered food, financial aid, COVID-19 safety guideline, and community-driven health interventions, including anti-retroviral therapy for sex workers during the COVID-19 pandemic[157,162]. Distribution of anti-retroviral therapy in private and discreet sites and utilisation of WhatsApp messaging to share information related to the pandemic were conducted[158]. <br>• Thailand: community-led organisations provided food, hand sanitisers, condoms and face masks to the sex workers[157]. <br>• Singapore: the pandemic and its lockdown measures increased the economic hardship of sex workers[161]. <br>• Africa: community-led outreach played a role in distributing food packs to sex workers[163]. |
| Victims of domestic violence | • Increased incidence of domestic violence. <br>• Difficulties in leaving home to access institutional help. | • Public information campaigns. <br>• Helpline and other online support services. <br>• Special allowance given to victims to bypass the lockdown restrictions and travel to sheltered accommodations. | • Argentina, Australia, Bangladesh, Brazil, China, Democratic Republic of Congo, France, Guatemala, India, Portland, South Africa, Spain, the UK, and the US had observed varying degrees of increase in domestic violence (e.g., against women or children) during the enforcement of physical distancing measures and lockdowns[164,167–172,175,176,233,234]. |

**Table 1 (continued) | Summary of findings across vulnerable populations**

| Vulnerable groups | Negative impacts of physical distancing | Ringfence measures | Exemplar countries |
|---|---|---|---|
| | | | • Bangladesh, South Africa, Spain: many channels to help the abused victims had shut down or slowed down in their operations during the pandemic[167,169,173].<br>• France and Italy: governments commissioned hotels as shelters for victims of domestic violence[164].<br>• UK: national campaigns were organised to raise domestic abuse awareness and highlighted available assistance services;[176] helpline services were extended for victims of domestic violence, and funds were allocated to support these helplines and other online support services;[176] victims of domestic violence were allowed to bypass the lockdown restrictions and travel to sheltered accommodations to seek help;[176] schools and universities also extended accommodations, counselling, and support to students vulnerable to domestic violence[176]. |
| Refugees | • Disruption of government services serving refugee camps.<br>• Lack of mental health support for refugees already living with post-traumatic stress disorders or other mental illnesses.<br>• Children in refugee shelters facing difficulties in participating in remote learning. | • Therapy over the phone or video conferencing.<br>• Voluntary delivery of mental health service to refugees.<br>• Offering face masks to refugees and educating them to use the masks.<br>• Detection and isolation of suspected cases. | • Refugees were usually excluded from financial and social support programmes (e.g., food relief programme) offered by the destination countries, such as in South Africa[208].<br>• Berlin, Germany: children in refugee shelters found it difficult to participate in remote learning due to the lack of laptops or internet connection[235].<br>• Boston, US: the Boston Center for Refugee Health and Human Rights offered teletherapy via phones or video conferencing with refugees[187].<br>• Italy: social workers helped to address the social and mental health vulnerabilities that asylum-seekers faced during the pandemic[189]. |
| Ethnic minorities | • Difficulty in following physical distancing due to overcrowded housing, adverse financial and employment conditions.<br>• Lack of translated or visually supportive materials for non-English speakers so that they could understand the physical distancing measures.<br>• Negative impacts on mental health. | • Dissemination of COVID-19 information effectively through translated and visualisation materials.<br>• Special education services offered for non-English language learners to facilitate effective remote learning. | • In some ethnic minority communities in the UK and US, households comprising multigenerational family members living in crowding housing conditions made adherence to physical distancing measures challenging[236,237].<br>• India: food rations to ethnic minority households because of a shortage of food[93].<br>• UK: non-English speaking Black and Asian ethnic minority groups lacked access to translated or visually supportive materials conveying public information about the pandemic from the government;[192,238] UK's lockdown deteriorated socio-economic positions of ethnic minorities and negatively affected their access to food and necessities[238].<br>• US: special education services to support remote learning for non-English speakers from kindergarten to 12th grade[75]. |
| People from sexual and gender minorities | • Disruption of access to essential medical services, including HIV testing and treatment.<br>• Negative impacts on mental health. | • Online delivery of telehealth interventions pertaining to HIV prevention and care services.<br>• NGO-led support in the forms of online chat, or online counselling services, or materials (e.g., free masks, food). | •. Brazil: transgender population, especially those who were older, reported significantly higher mental health issues and lack of access to healthcare during the pandemic[199].<br>• UK and US: providing telehealth interventions such as web and text-based chats and online video/audio counselling services for the people from sexual and gender minorities[205–207].<br>• India and US: offering temporary housing, financial aid, food, and free masks to the people from sexual and gender minorities[205,239]. |

accessible to socially isolated population groups, such as people in prison and ethnic minorities. For children/students, school closures lead to concerns about disruption to their learning and decreased social interactions with peers, especially for children from low-income families. For low-income populations and migrant workers, job losses and financial challenges are the most common challenges faced during lockdowns. Lockdowns or stay-at-home orders also hamper access to social support among refugees and people who are at high risk of domestic violence.

Various ringfenced measures have been taken by governments to address the diverse challenges faced by vulnerable populations and these have been discussed by scholars; they range from the use of technologies to telehealth services and financial support. To date, there has been limited evidence on the impacts of these measures. Evaluating these policy measures represents an important future research direction and is critical to inform policymakers, so that they can adopt an optimal set of ringfenced policy measures to enhance social inclusion for vulnerable populations.

Among the countries that have implemented physical distancing measures, few have ringfenced measures for vulnerable groups (e.g., income support for the low-income populations), which suffer from the unintended consequences of physical distancing[204]. Effective control of the COVID-19 pandemic requires packaging policy measures strategically to address multiple competing objectives and to balance

**Table 2 | Different vulnerable groups and their vulnerabilities, as discussed in the literature**

| Vulnerable group | Primary vulnerability | Secondary vulnerabilities |
|---|---|---|
| Older people | Health vulnerability<br>• High risks of severe COVID-19 symptoms or COVID-19 complications. | Social vulnerability<br>• Overcrowded accommodation in some LTC facilities. |
| Children/students | Cognitive or communicative vulnerability<br>• Insufficient ability to comprehend information and make decisions. | Deferential vulnerability<br>• Their decisions/behaviours are under the influence/control of, or obligated to, third parties. |
| Low-income populations | Economic vulnerability<br>• Adverse employment conditions (unemployment, working on temporary contracts, part-time employment, self-employment, informal sector workers).<br>• Adverse financial conditions (payment arrears, or low-income).<br>• Digital and connectivity conditions (lack of access to internet, or lack of access to information technology). | Social vulnerability<br>• Poor-quality housing (lack of access to basic necessities, e.g., water and sanitation).<br>• Overcrowding housing.<br>• Socially isolated. |
| Migrant workers | Economic vulnerability<br>Adverse employment conditions.<br>Adverse financial conditions. | Social vulnerability<br>Poor-quality housing (lack of access to basic necessities, e.g., water and sanitation).<br>Overcrowding housing.<br>Socially isolated. |
| People in prison | Institutional or deferential vulnerability<br>• Deprived of liberty, and kept in prison under the control of prison wardens. | Social vulnerability<br>• Overcrowding housing.<br>• Socially isolated. |
| People with disabilities | Health vulnerability<br>• Long-term sick or disabled.<br>• Terminally ill individuals. | Cognitive or communicative vulnerability<br>• Impaired decision-making ability.<br>Economic vulnerability<br>• Dependent.<br>• Adverse employment conditions.<br>• Adverse financial conditions. |
| Sex workers | Health vulnerability<br>• Vulnerable to sexually transmitted infections. | Economic vulnerability<br>• Adverse financial conditions (payment arrears or low income).<br>Social vulnerability<br>• Vulnerable to violence. |
| Victims of domestic violence | Social vulnerability<br>• Socially isolated.<br>• Vulnerable to violence. | Deferential vulnerability<br>• Decisions/behaviours under third-party influence/control.<br>Economic vulnerability<br>• Dependent. |
| Refugees | Social vulnerability<br>• Socially isolated.<br>• Vulnerable to violence.<br>• Crowded informal refugee camps with a lack of water and sanitation facilities. | Economic vulnerability<br>• Adverse employment conditions.<br>• Adverse financial conditions.<br>Health vulnerability<br>• Vulnerable to mental health problems. |
| Ethnic minorities | Social vulnerability<br>• Socially isolated/marginalised. | Economic vulnerability<br>• Members of the ethnic minority were more likely to live in poverty than the ethnic majority, on average, and less likely to work in high or semi-skilled jobs. |
| People from sexual and gender minorities | Health vulnerability<br>• Vulnerable to HIV risks.<br>• Vulnerable to mental health illness. | Social vulnerability<br>• Socially isolated/marginalised.<br>• Vulnerable to violence. |

the various conflicts and tensions that exist between controlling the virus's transmission to save lives, supporting the economy and ensuring that the wellbeing of vulnerable groups is accounted for in policy processes. Physical distancing measures should be complemented by ringfenced measures to reduce the negative impacts of physical distancing measures on vulnerable groups and to enhance the overall policy effectiveness. More research is required to weigh the trade-offs between the benefits and unintended negative consequences of physical distancing measures, to improve their design and to minimise long-term socio-economic disparities.

Among studies of the experience of vulnerable groups during the COVID-19 pandemic, most studies focus on low-income populations, the older people and children/students, while other vulnerable groups receive much less attention. Each group may experience multiple forms of vulnerability that are mutually reinforcing. Here we identify their primary and secondary vulnerabilities (Table 2).

To safeguard vulnerable populations and meet their financial, health, education, food and housing needs during the COVID-19 pandemic, the following strategies have gained traction in the literature,

that while speculative at this stage, merit policy considerations: (a) Leveraging technological prowess of autonomous systems and digital technologies through use of technologies such as social companion robots, to address social isolations faced by the older residents in care homes[22–25] and ICT to deliver health services, such as online counselling services by mental health professionals[10,22–25,132], (b) Ensuring continuity of essential health services[129,132,157,162] and strengthening health service delivery capacity and use of telemedicine or home care services, to ensure treatment continuity for patients[205–207], (c) Providing financial support for low-income populations in the forms of direct cash assistance, in-kind assistance, low-interest loans, tax reduction or rebates, and temporary relief funds or concessions for small businesses, and flexible payment options for utilities and other essential public services[35,87,93,96,99,113,114], as well as rolling-out various unemployment benefits such as handouts or improving employability via upskilling and retraining[79,87,115], (d) Establishing effective public communication for vulnerable populations who are socially isolated[132,139,158,192] and a comprehensive support package for them to ensure reliable access to necessities, such as food,

**Table 3 | Categorisation of vulnerabilities and vulnerable groups in this research**

| Sources | Categorisation of vulnerabilities and vulnerable groups |
|---|---|
| NBAC[210], Yale University[211], OECD[212], Mikolai et al.[213], and Mishra et al.[214] | (1) Cognitive or communicative vulnerability, e.g., children, decisionally impaired persons, people struggling with low subjective wellbeing or poor mental health conditions. |
| | (2) Institutional or deferential vulnerability, e.g., people in prison, or children/students. |
| | (3) Health vulnerability, e.g., long-term sick or people with disabilities, terminally ill subjects, seriously ill subjects, those with high risk of exposure to the virus, high risk of severe COVID-19 symptoms or COVID-19 complications (the older people, people with chronic diseases). |
| | (4) Economic vulnerability, e.g., dependent persons, or impoverished people; those with adverse employment conditions (unemployment, working on temporary contracts, part-time employment, self-employment, informal sector workers, or migrant workers); adverse financial conditions (payment arrears, or low-income); digital and connectivity conditions (lack of access to internet, or lack of access to PC/laptop/tablet/netbook). |
| | (5) Social vulnerability, e.g., poor-quality housing (lack of access to necessities, e.g., water and sanitation); overcrowded housing; the socially isolated; vulnerability to violence (sex workers, victims of domestic violence). |

housing/shelter, electricity, water and sanitation[85,93,104,106,157,162,208], (e) Enhancing the capacity of the education sector to deliver online or home-based learning[31–37,75,76] and outreach services to support special needs students with learning difficulties[153], and (f) Strengthening public finance to facilitate implementation of physical distancing measures[35,39,73,74,176] and to weather the economic recession impacts caused by future pandemics[35,73,74].

This scoping review has two potential limitations. First, we may have missed out a small subset of studies during the systematic evidence search process due to the various ways vulnerabilities and vulnerable populations are defined and conceptualised. Second, we did not attempt to tease out the jurisdictional differences pertaining to the negative impacts experienced by the 11 vulnerable populations examined in this review. This presents an opportunity for future research.

### Broader impact

This scoping review represents a comprehensive effort to consolidate empirical insights regarding the impacts of COVID-19 physical distancing measures on the most vulnerable in society, and to identify strategies and actions that have been taken to address their vulnerabilities. Review studies have been conducted on physical distancing measures and their effectiveness[3–5], but few reviews have focused specifically on vulnerable populations. By examining the extent to which universally applied physical distancing measures have negatively impacted the vulnerable populations, this scoping review fills a pertinent research gap and has significant policy implications for equity and human rights.

Our findings emphasise the need for policymakers and practitioners to pay more attention to addressing the needs and improving the welfare of the vulnerable populations as the world transits into endemicity. Furthermore, the insights and recommendations provided in this research would allow countries to put more efforts and resources into strengthening health provisions and social safety nets for vulnerable populations to better prepare for future public health emergencies.

In summary, this scoping review has identified more discussions of problems than solutions as regards addressing the needs of vulnerable population groups when imposing physical distancing policy measures to control the spread of COVID-19. The negative impacts of physical distancing measures for multiple vulnerable population groups include unemployment and income loss, prolonged social isolation (leading to loneliness and an increased mental health burden), as well as disruption to access to non-COVID-19-related health services and delayed treatment, such as for non-communicable diseases for the older people and sexually transmitted diseases for sex workers and the people from sexual and gender minorities.

We also find that physical distancing measures exacerbate the vulnerabilities of different vulnerable population groups. Adverse employment and financial conditions constitute the primary vulnerability of low-income populations and migrant workers. Health vulnerability is typical among the older people and the people with long-term disabilities. Sex workers and the people from sexual and gender minorities are more vulnerable to sexually transmitted diseases. Social vulnerability is the primary vulnerability faced by victims of domestic violence, refugees and ethnic minority population groups. Regarding children, cognitive/communicative vulnerability is a major concern, and they need help to keep up with online learning. People in prison primarily face institutional/deferential vulnerability, as their decisions and behaviour are supervised by prison wardens.

The aggravation of the state of vulnerability of these populations, which were already vulnerable prior to the COVID-19 pandemic, should not be overlooked by governments. Addressing these negative impacts is of paramount importance to many countries as they have ripple impacts that affect the larger general population. Ignoring or downplaying the plight of vulnerable populations affected by the COVID-19 pandemic can translate into higher social care and financial burdens for governments in future as the pandemic continues to ravage the global economy.

## Methods
### Types and descriptions of vulnerability
While the meaning of "vulnerability" may be defined somewhat differently in various fields, the term "vulnerable populations" commonly refers to social groups that are exposed to increased risks or susceptibility to adverse health outcomes or diminished quality of life[209]. Based on the categorisation of the National Bioethics Advisory Committee of the United States (NBAC)[210] and Yale University[211], and drawing insights from the Organisation for Economic Co-operation and Development (OECD)[212], Mikolai et al.[213] and Mishra et al.[214] (Supplementary Text A2), we constructed the categorisation of vulnerabilities and vulnerable groups shown in Table 3. It should be noted that these categories of vulnerable groups or vulnerabilities are not mutually exclusive. Different types of vulnerability may intersect with one another. For instance, while people in prison face institutional or deferential vulnerability, since they are incarcerated, they are also subject to overcrowding and are vulnerable to violence, under the social vulnerability category. Lack of access to necessities (e.g., water and sanitation) or lack of access to information technology can be co-producers of social and economic vulnerabilities.

### Study design
A scoping review was conducted to examine the types and nature of physical distancing measures implemented across the world, and the extent to which they have negatively impacted vulnerable populations.

**Table 4 | Key terms included in search strings**

| Concepts | Key terms in search strings |
|---|---|
| Covid-19 | "ncov" OR "2019 ncov" OR "Covid-19" OR "Covid19" OR "Covid-2019" OR "Covid2019" OR "sars-cov-2" OR "sars cov-2" OR "sarscov2" OR "sarscov-2" OR "sars-coronavirus-2" OR "sars corona virus" OR "sars-like coronavirus" OR "novel coronavirus" OR "novel corona virus" OR "Covid*" OR "coronavirus 2" OR "coronavirus infection*" OR "coronavirus disease" OR "corona virus disease" OR "new coronavirus" OR "new corona virus" OR "new coronaviruses" OR "novel coronaviruses" OR "severe acute respiratory syndrome coronavirus 2" OR "coronavirus" OR "sars-cov" |
| Physical distancing | "social distancing" OR "social isolation" OR "physical distancing" OR "physical distance" OR "safe distancing" OR "lockdown" OR "lock down" OR "quarantine" OR "stay-at-home" OR "stay at home" OR "self isolation" OR "self-isolation" OR "remote work" OR "school closure" OR "workplace closure" |
| Policy measure | "act" OR "design" OR "govern*" OR "intervention" OR "law" OR "legislation" OR "politics" OR "regulation" OR "policy" OR "policies" OR "policy measure" OR "policy instrument" OR "policy mix" OR "policy bundle" OR "policy package" |

**Table 5 | Inclusion and exclusion criteria for screening of relevant studies**

| Inclusion criteria |
|---|
| (1) Studies examining vulnerable populations as populations of interest. |
| (2) Studies examining various physical distancing measures from the public policy and/or legal perspectives. |
| (3) Peer-reviewed studies (empirical, conceptual and review studies), policy briefs, reports, editorials, commentaries, perspectives and letters. |
| (4) Studies employing jurisdictions (prefecture/district/country/state/province, single country, multi-countries) as a unit of analysis. |
| (5) Studies employing quantitative, qualitative or mixed-methods research designs. |
| (6) Studies published as full-text articles in the English language between November 2019 and June 2022. |

| Exclusion criteria |
|---|
| (1) Studies examining physical distancing measures but that do not mention their impacts on vulnerable populations. |
| (2) Clinical studies on the COVID-19 pandemic without public policy and/or law dimensions. |
| (3) Studies published before November 2019. |
| (4) Studies for which full-text articles are not accessible or that are not published in the English language. |

A consolidation of global evidence is timely due to the rapid emergence of evidence since the beginning of the pandemic, and the lack of focus on the invisible and hidden impacts of physical distancing interventions—which have been applied nearly universally—on vulnerable populations that are already prone to many disadvantages due to pre-existing socio-economic fault lines.

In this scoping review, we posed the following review questions: (i) What physical distancing measures or interventions have been implemented and have they negatively impacted vulnerable populations? (ii) What ringfenced measures have been designed to protect vulnerable populations during the COVID-19 pandemic and how have they been implemented?

### Search strategy and data sources

From March 2021 to April 2021, we searched ten databases to identify articles that could potentially be relevant to the issue of physical distancing measures. The databases were PubMed, Scopus, Web of Science, ProQuest, ProQuest Coronavirus Research Database, Embase, Educational Resource Information Center database, LITCOVID, the Cochrane database of systematic reviews, and WHO's database of global literature on coronavirus disease. We developed the search strategy and search strings (Table 4) with the help of an experienced information specialist. The key terms used for the literature search focused on the themes of COVID-19 and physical distancing policy measures.

During the peer review process, in June 2022 we updated the search to include new literature published after April 2021 for vulnerable populations (the strategy for the updated search is in Supplementary Text A3). We removed duplicate results from the updated searches against previous results to ensure records were accurate.

### Eligibility criteria and screening processes

Table 5 summarises the eligibility criteria for the screening of the studies.

The screening process involved screening the titles and abstracts to select articles relevant to vulnerable population groups, following the full inclusion and exclusion criteria and referring to the categories of vulnerabilities listed in Table 3. The first author conducted the entire screening process independently to identify relevant studies, while the third author randomly selected more than half of the records to screen independently. Both authors went through two iterations to achieve less than 10% discrepancies. All discrepancies were resolved through a detailed discussion among all three authors. Thereafter, the third author again cross-checked the records chosen by the first author to achieve final agreement on the included studies. Full texts of the identified relevant articles were later retrieved for data extraction.

### Data extraction

Data extraction followed a predesigned data extraction template created through ongoing discussion among all the authors. To ensure quality control, all three authors piloted the data extraction practices for the first 10% of the identified full-text articles. The authors then discussed the data extraction results to build a consistent understanding of the aim and scope of the review. Thereafter, the first author extracted data from the remaining articles, and the other two authors validated a random selection of the data extraction results to ensure consistency.

### Data analysis

We managed extracted data using Excel 16 (Microsoft Corporation). The data synthesis involved intensive line-by-line reading of the extracted qualitative information by all the authors. We grouped the studies by categories of vulnerable population groups examined using a framework synthesis approach. Framework synthesis enables structured analysis to be done by following a five-stage approach (familiarisation of the issue, framework selection, indexing, charting and finally mapping and interpretation). It is a versatile analytical approach that accounts for heterogeneity in the types of study included

(quantitative, qualitative and mixed method) and is a suitable approach when theory is nascent and emergent, which was the case in this review[215]. At every stage of the analysis, discussions were held to achieve final agreement on the results.

## Reporting summary

Further information on research design is available in the Nature Portfolio Reporting Summary linked to this article.

## Data availability

The authors declare that all data generated and analysed during the current research are included in this article. The studies included in this scoping review are listed in its supplementary files.

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

## Acknowledgements

This research is supported by Ministry of Education Singapore AcRF Tier 1 funding support through NUS ODPRT Reimagine Grant and the Lee Kuan Yew School of Public Policy, National University of Singapore with grant numbers A-0003291-01-00 (A.T.) and A-0003291-00-00 (A.T.). A.T. is grateful for the support provided by the NUS ODPRT Reimagine Research grant scheme and encouragements of Prof CHEN Tsuhan.

## Author contributions

Conceptualisation, A.T.; methodology, A.T. and S.-Y.T.; validation, A.T.; formal analysis, L.L., A.T. and S.-Y.T.; investigation, L.L., A.T. and S.-Y.T.; resources, A.T.; data curation: L.L. and S.-Y.T.; writing and editing L.L., A.T. and S.-Y.T.; supervision, A.T.; project administration, A.T.; funding acquisition, A.T. All authors read and agreed to the published version of the manuscript.

## Competing interests

The authors declare no competing interests.
