## [Peer Review File · Nature Communications]

A scoping review of the impacts of COVID-19 physical distancing measures on vulnerable population groupsREVIEWER COMMENTS

Reviewer #1 (Remarks to the Author):

The article "A systematic review of the impact of social distancing measures on vulnerable population groups" by Lili Li and colleagues is a heartbreaking synthesis of literature on the disruptions to the lives of the vulnerable in the wake of COVID-19 control measures. This is a monumental work of documentation, and the nature of the material – and its breadth – make a systematic review the proper approach (as opposed to, say, a meta-analysis).

The authors systematically identify vulnerable groups using established criteria. They then go through the literature to extract and review 179 articles documenting both impacts as well as suggested mitigation approaches for 11 broad groups of vulnerable populations. It is a moving and compassionate review of many issues that were shelved while efforts to contain COVID-19 took center stage.

The authors write with great compassion for the many efforts to contain COVID-19, and their focus on vulnerable groups is thoughtful throughout.

This article combines a thorough and systematic account of the literature with a stirring and painful but sober look at the most vulnerable. It deserves to be published.

A few specific comments, largely about the writing:

1. The review of the literature was done from March to April 2021. Much more has been documented since then. I would not hold an update against publication, since the effort that went into the current review is very substantive. However, any updates would only strengthen the article.
2. There is some jumping around in each of the sections 4.2-4.12. It may be helpful to organize each paragraph in the same order of topics. For example: group vulnerability; vulnerability to COVID-19; impacts of social distancing measures; mitigation strategies for unintended harms. Whether or not these (or similar) sub-headings make it into each section, following a parallel structure will make the findings easier to follow.
3. There is a lot of passive voice. For example, "it was suggested" or "services were provided." Switching to active voice ("some studies suggest that...") would strengthen the article.
4. Whenever possible, adding specific examples from specific article would strengthen the narrative.

Overall, this is an impressive and important contribution. It is so important to document and summarize these issues as we start to recover from the past two years. And this article takes a compassionate, balanced, and thorough look at those with the greatest vulnerabilities and fewest voices.

Reviewer #2 (Remarks to the Author):

General

This is a relevant topic, reviewing existing literature on negative effects of physical distancing measures on vulnerable populations. It is not bad, but there are several problems with the manuscript that could be difficult to correct without redoing aspects of your search, including lack of definitional clarity on key concepts, lack of justification/clarity in methods, and overreliance on appendices. This could have been conducted as a nice scoping review, given the broad question, but instead seems to be a scoping-systematic hybrid that satisfies criteria for neither type of review.

Title

1. Ensure your title is accurate (e.g. you only looked at negative effects) and includes the geographic scope of your topic (e.g. global or which region/countries).

Abstract/key words

2. Clarify how you are defining 'vulnerable groups' and 'social distancing' when you first use these terms.
3. Add another sentence to clarify search, screening, extraction, and analysis, including what you mean by 'systematic synthesis' and defining you 11 population groups and 6 recommendations (or not specifying numbers).
4. Keywords: Are 'policy measure,' 'unintended consequence' or 'ringfenced measure' likely to be useful search terms? I doubt it and suggest deleting/replacing.

Introduction

5. Lns 35-36/60, 86 etc, since I don't have any supplemental material or appendices to hand, don't rely on these or expect readers to go searching. Include anything important for reader understanding in the manuscript body.
6. Since you're actually talking about 'physical distancing' or 'safe distancing', better to use one of these more accurate terms instead of the misnomer 'social distancing'.
7. 'Vulnerable' is a problematic term, so please at least briefly define how you will be using it and why at first use.
8. Clearer justification is needed beyond your apparent decision to do this review because someone else had already reviewed effectiveness.
9. For public health research, put your aim and objectives at the end of your introduction and move your research question/s to Methods. Regarding your questions, why did you only look at negative affects and ringfencing and how are you defining ringfencing?
10. Conceptualisation: good to include this, but it is more a description of your chosen definition than a theorisation or conceptualisation. So, correct your terminology or revise this and in either case it belongs in the first sub-section of your Methods

Methods

11. This actually reads like a hybrid scoping review/systematic review and you would have been better off doing a scoping review. So, overall this section needs a lot of work and should be restructured roughly as follows:
12. First sub-section (Study design) should identify the type of review and why chosen, any theory/conceptualisation, study definitions (e.g. table 1 including all, not just vulnerabilities), and research questions.
13. Second sub-section (e.g. Search strategy/Source identification) should justify the databases searched as well as listing them, and provide an example search syntax as narrative or table (don't send readers to another appendix).
14. Third sub-section (Screening) should include a table of your eligibility criteria and explain how you conducted double screening of both title/abstracts and full texts. As I don't know your eligibility criteria, I can't judge whether they are reasonable. Additionally, more detail is needed as it's unclear to me how you shared out title/abstract screening, if/how you screened full texts, and how your <10% discrepancy was estimated. This appears to have used a convenience rather than systematic approach.
15. Fourth sub-section (Quality assessment) is missing and should be included.
16. Fifth sub-section (Data extraction) should list your extraction headings and justify your approach to dividing the task between authors, as again this does not appear systematic.
17. Sixth sub-section (Analysis) should identify and justify your analysis or synthesis method (e.g. line-by-line reading is not a type of analysis) and clarify who did what.

Findings

18. Such a large number of duplicate and irrelevant sources suggests issues with your database selection and search syntax. However, I can't judge these as they aren't included. Your screening process (which should anyway be in Methods) is unclear. Did you do an initial title-only screening and if so, why? First should be title/abstract, second should be full text screening. Please clarify.
19. Fig2 is too difficult to read. Please revise.
20. Findings section needs some clarification. Please restructure sub-sections in alignment with your synthesis/analysis (which you will have explained in Methods). Ideally, give an indication in each sub-section of how many sources discussed each topic and how social groups were defined. Your findings are entirely descriptive - which isn't problematic in itself - but it's therefore even

more difficult to determine how/if you prioritised and structured your findings - this is unfortunate, as the information you provide is interesting.

Discussion

21. Your paragraph on your study's contributions needs more detail, e.g. it should include whether anything in your findings is new or if it reinforces existing knowledge.

22. Don't include new data or data tables in your Discussion - these should be in Findings

23. Include a limitations sub-section

24. Include a sub-section on how study findings relate to existing knowledge and general relevance for policy, practice and further research as otherwise your recommendations table comes out of nowhere. Additionally, Table 4 seems more like a list of preferences/wishes rather than being prioritised by feasibility or value or grounded in evidence.

Level of interest: An article of relevance to those with similar interests

Quality of written English: Good

Statistical review: No

Response to Referees Letter

Revision - NCOMMS-22-08041-A A systematic review of the impact of social distancing measures on vulnerable population groups

Responses to the reviewers,

We would like to thank the two reviewers for their time, helpful comments, and suggestions. We were pleased that we have been given the opportunity to further refine the manuscript, and that, in the words of reviewer 1 “This article combines a thorough and systematic account of the literature with a stirring and painful but sober look at the most vulnerable. It deserves to be published”, and in words of reviewer 2 for addressing a relevant topic research.

All comments are appreciated and below we describe how we accounted for each of the comments and, overall, how we improved the quality of the manuscript and clarified its limitations. We have made extensive changes that are highlighted using tracked changes in the revised manuscript and/or below.

Reviewer# 1

The article “A systematic review of the impact of social distancing measures on vulnerable population groups” by Lili Li and colleagues is a heartbreaking synthesis of literature on the disruptions to the lives of the vulnerable in the wake of COVID-19 control measures. This is a monumental work of documentation, and the nature of the material – and its breadth – make a systematic review the proper approach (as opposed to, say, a meta-analysis).

The authors systematically identify vulnerable groups using established criteria. They then go through the literature to extract and review 179 articles documenting both impacts as well as suggested mitigation approaches for 11 broad groups of vulnerable populations. It is a moving and compassionate review of many issues that were shelved while efforts to contain COVID-19 took center stage.

The authors write with great compassion for the many efforts to contain COVID-19, and their focus on vulnerable groups is thoughtful throughout.

This article combines a thorough and systematic account of the literature with a stirring and painful but sober look at the most vulnerable. It deserves to be published.

A few specific comments, largely about the writing:

1. The review of the literature was done from March to April 2021. Much more has been documented since then. I would not hold an update against publication, since the effort that went into the current review is very substantive. However, any updates would only strengthen the article.

Response: Thank you for your comment. We agree with the reviewer that data updates would strengthen the article and we have taken substantive efforts to perform an update of the systematic review search over the past three months. Specifically, we conducted a targeted and extensive search to identify new and relevant articles published since April 2021. Our search resulted in 7989 hits. We screened these 7989 articles based on the same set of

inclusion and exclusion criteria and identified additional 87 studies that are relevant to the review. We extracted the data for these additional 87 studies and analysed their findings. With the addition of these new studies, we did not find major deviations from our previous findings. We added the description of the updated search to the manuscript and updated the findings section accordingly. List of added studies is also updated in Supplementary Table 1~Table 2. Essentially, findings from the updated search did not fundamentally change our original findings.

[Method section] During the peer review process, in June 2022 we updated the search to include new literature published after April 2021 for vulnerable populations (the strategy for the updated search is in Supplemental Text A3). We removed duplicate results from the updated searches against previous results to ensure records were accurate.

[Findings section] Of the total 39,816 records that were produced by searches, we identified 265 eligible studies for synthesis of their results in this review.

2. There is some jumping around in each of the sections 4.2-4.12. It may be helpful to organize each paragraph in the same order of topics. For example: group vulnerability; vulnerability to COVID-19; impacts of social distancing measures; mitigation strategies for unintended harms. Whether or not these (or similar) sub-headings make it into each section, following a parallel structure will make the findings easier to follow.

Response: Thank you for the suggestion. Accordingly, we re-organized each section following the order of group vulnerability, vulnerability to COVID-19, impacts of social distancing measures, and mitigation strategies for unintended harms. Table 1 (i.e., the old Table 2) also structurally shows information about the impacts of social distancing measures and mitigation strategies for unintended harms relating to each vulnerable population.

3. There is a lot of passive voice. For example, “it was suggested” or “services were provided.” Switching to active voice (“some studies suggest that...”) would strengthen the article.

Response: Thank you for your comment. Following your suggestion, we have substantially revised the manuscript to enhance the language and have also specifically switched sentences in passive voices into active voices. We also used the services of a professional copyeditor from Oxford to further improve the quality of the language.

4. Whenever possible, adding specific examples from specific article would strengthen the narrative.

Response: Thank you for your suggestion. We have added examples from specific articles throughout the manuscript, particularly in Finding Section. Table 1 (i.e., the old Table 2) also summarises specific country cases discussed in the literature. We also include a more comprehensive narrative in the Supplementary Information.

Overall, this is an impressive and important contribution. It is so important to document and summarize these issues as we start to recover from the past two years. And this article takes a compassionate, balanced, and thorough look at those with the greatest vulnerabilities and fewest voices.

Response: We thank reviewer 1 for acknowledging our systematic review to be an impressive and important contribution to the literature, and for the positive feedback on our choice to scope this review to focus on vulnerable populations who are experiencing the greatest vulnerabilities with fewest voices during this pandemic.

Reviewer# 2

General

This is a relevant topic, reviewing existing literature on negative effects of physical distancing measures on vulnerable populations. It is not bad, but there are several problems with the manuscript that could be difficult to correct without redoing aspects of your search, including lack of definitional clarity on key concepts, lack of justification/clarity in methods, and overreliance on appendices. This could have been conducted as a nice scoping review, given the broad question, but instead seems to be a scoping-systematic hybrid that satisfies criteria for neither type of review.

Response: We thank reviewer 2 for appreciating that we address a relevant topic in this systematic review. We also acknowledge reviewer 2's concerns on certain areas of the review, including conceptual clarity, methods clarity and overreliance on appendices. In the following responses, we document how we have accounted for all the comments raised and made efforts to strengthen the conceptual clarity, justify the need for a systematic review in conducting our research, and incorporate many details embedded in the appendices earlier to the main text of the paper.

Specifically, regarding the choice of methods, we retain the use of a systematic review. This is because we aim for depth, contexts and nuances in our findings and a systematic review is a better methodological approach to achieve this as compared to a scoping review (Munn et al. 2018). In addition, we have adhered to the rigorous PRISMA guidelines for a systematic review, which include developing a search string, systematically screening for studies that fulfil the inclusion and exclusion criteria, extracting the data for all included studies, critically appraising all the included studies (incorporating the reviewer's comment in Q.15 below) and synthesising the findings using a line-by-line reading of the extracted qualitative information.

Reference:

Munn Z, Peters MDJ, Stern C, Tufanaru C, McArthur A, Aromataris E. Systematic review or scoping review? Guidance for authors when choosing between a systematic or scoping review approach. *BMC Med Res Methodol.* 2018 Nov 19;18(1):143. Doi: 10.1186/s12874-018-0611-x. PMID: 30453902; PMCID: PMC6245623.

Title

1. Ensure your title is accurate (e.g. you only looked at negative effects) and includes the geographic scope of your topic (e.g. global or which region/countries).

Response: Thanks for this comment. Based on your suggestion we have revised the title from: "A systematic review of the impact of social distancing measures on vulnerable population groups" to "Revised title: A systematic review of the negative effects of physical distancing measures on vulnerable population groups worldwide"

Abstract/key words

2. Clarify how you are defining ‘vulnerable groups’ and ‘social distancing’ when you first use these terms.

Response: Thank you for this comment. We had defined the terms in detail in the supplemental information, following your suggestion, we added definitions of ‘vulnerable groups’ and ‘social distancing’ to the abstract when we first used the terms.

Abstract: Most governments have enacted physical distancing measures that controlled the Covid-19 transmission by maintaining a physical distance between people and reducing crowds. Yet, little is known about the socio-economic trade-offs of these measures, especially for vulnerable groups exposed to increased risks or susceptibility to adverse health outcomes or diminished quality of life.

We have now provided more detailed explanations of these terms in Section 1 ‘Introduction’, in Section ‘Conceptualisation of vulnerable populations and their vulnerability’, and in the Supplemental File.

Section 1 ‘Introduction’ has the following elaboration of ‘**social distancing**’.

One widely implemented tool in governments’ arsenals that has been used to curb the spread of Covid-19 is the deployment of “physical distancing” (often used interchangeably with the term “social distancing”) measures. According to the World Health Organization (WHO), social distancing aims to maintain safe physical distancing through decreased crowding.² Social distancing (hereafter physical distancing) measures range from lockdowns and school closures to restrictions on social gatherings in homes and public places (Supplemental Text A1).

Supplementary Text A1 added the examples of social distancing measures as below.

Supplementary Figure 1. Social distancing measures listed by WHO²

The Section ‘Conceptualisation of vulnerable populations and their vulnerability’ elaborates different **vulnerable populations and vulnerabilities**.

The categorisation of vulnerabilities and vulnerable groups (see Table 4) were established based on existing concepts in the literature.

Based on the categorisation of the National Bioethics Advisory Committee of the United States (NBAC)²³⁷ and Yale University,²³⁸ and drawing insights from the Organisation for Economic Co-operation and Development (OECD),²³⁹ Mikolai et al.²⁴⁰ and Mishra et al.²⁴¹ (Supplemental Text A2), we constructed the categorisation of vulnerabilities and vulnerable groups shown in Table 4.

Supplementary Text A2 describes the vulnerability categorisation of National Bioethics Advisory Committee and Yale University, and insights from the Organisation for Economic Co-operation and Development, Mikolai et al. and Mishra et al.

References:

237. NBAC (National Bioethics Advisory Commission of the United States). *Ethical and policy issues in research involving human participants. Report and Recommendations of the National Bioethics Advisory Commission* (2001).
238. Yale University. Categories of vulnerability – specific vulnerable populations. *HSP Module 7: Protecting Vulnerable Subjects* Available at: <https://assessment-module.yale.edu/human-subjects-protection/categories-vulnerability-specific-vulnerable-populations>. (Accessed: 27th July 2021)
239. OECD. *COVID-19: Protecting people and societies. OECD Policy Responses to Coronavirus (COVID-19)* (OECD, 2020). doi:10.1787/9870c393-en
240. Mikolai, J., Keenan, K. & Kulu, H. Intersecting household-level health and socio-economic vulnerabilities and the COVID-19 crisis: An analysis from the UK. *SSM - Popul. Heal.* **12**, (2020).
241. Mishra, S. V., Gayen, A. & Haque, S. M. COVID-19 and urban vulnerability in India. *Habitat Int.* **103**, (2020).

3. Add another sentence to clarify search, screening, extraction, and analysis, including what you mean by ‘systematic synthesis’ and defining you 11 population groups and 6 recommendations (or not specifying numbers).

Response: Thanks for this comment. Following your suggestion, we have added a sentence to clarify the search, screening, extraction and analysis process in the abstract. We also revised the last sentence in the abstract to clarify the six policy recommendations explicitly.

The sentence to clarify the search, screening, extraction and analysis process reads as follows: *“Based on ten inclusion and exclusion criteria, we screened 39816 records from major academic databases and synthesised results from 265 studies globally documenting the negative effects of physical distancing on 11 vulnerable population groups (elderly, children/students, low-income population, migrant workers, prisoners, disabled persons, sex workers, victims of domestic violence, refugees, ethnic minorities and LGBTQ+)”*

The last sentence reporting the six policy recommendations reads as follows:

We propose six policy recommendations to protect the vulnerable populations from the negative repercussions of the pandemic, centring on (i) leveraging digital technology, (ii)

ensuring continuity of essential health services, (iii) providing financial support and retraining/upskilling opportunities, (iv) establishing effective public communication, (v) enhancing capacity to deliver online-based learning for the schools, and (vi) strengthening public finance.

4. Keywords: Are ‘policy measure,’ ‘unintended consequence’ or ‘ringfenced measure’ likely to be useful search terms? I doubt it and suggest deleting/replacing.

Response: Thank you for your comment. Following raising this issue, we have reconsidered the keywords and revised the keywords to the following:

Covid-19, systematic review, physical distancing, social distancing, policy, vulnerable populations

Introduction

5. Lns 35-36/60, 86 etc, since I don’t have any supplemental material or appendices to hand, don’t rely on these or expect readers to go searching. Include anything important for reader understanding in the manuscript body.

Response: Thank you for your suggestion. Following your comment we have added a few more sentences/terms to the introduction section from the supplemental material to better to establish the context and rationale of this systematic review given the word limit. We highlight to the readers that the full contextual details can be found in the Supplementary Information.

6. Since you’re actually talking about ‘physical distancing’ or ‘safe distancing’, better to use one of these more accurate terms instead of the misnomer ‘social distancing’.

Response: Thank you for raising this issue. We used the term ‘social distancing’ in this systematic review as social distancing is a term most used by the World Health Organization (WHO) throughout the pandemic. However, you have raised a valid point, while ‘social distancing’ is a more encompassing term which also includes ‘physical distancing’ and ‘safe distancing’, a recent study has highlighted that the term ‘social distancing’ can have ‘unintended but detrimental effects, as it evokes negative feelings of being ignored, unwelcome, left alone with one’s own fears, and even excluded from society’ (Wasserman et al. 2020, p.1).

On this ground, we have changed ‘social distancing’ term to ‘physical distancing’ throughout the paper. We have changed our title from ‘social distancing’ to ‘physical distancing’, we also explained in the ‘Introduction’ section that these two terms are often used interchangeably. Thereafter, we have changed all the terms ‘social distancing’ to ‘physical distancing’.

Reference:

Wasserman D, van der Gaag R, Wise J (2020). The term “physical distancing” is recommended rather than “social distancing” during the COVID-19 pandemic for reducing feelings of rejection among people with mental health problems. *European Psychiatry*, 63(1), e52, 1–2 <https://doi.org/10.1192/j.eurpsy.2020.60>

7. 'Vulnerable' is a problematic term, so please at least briefly define how you will be using it and why at first use.

Response: Thank you for this comment. We acknowledge your concern, but the term 'vulnerable population' is commonly used in the literature during the Covid-19 pandemic, referring to populations who possess higher risk of being predisposed to Covid-19 infection due to the pre-existing socio-economic fault-lines that occur even before the pandemic. Even multilaterals such as the WHO and The World Bank used this term in their reports and briefs (WHO 2021; The World Bank, 2021). The WHO, in particular, suggests that vulnerable populations are '*individuals or groups experiencing multiple vulnerabilities that compound the barriers and impacts they face*'. (WHP 2021, p.1).

To align with your second question raised above, we would like to reiterate that we defined this term in the methods section and followed the paper format of the journal.

References:

WHO (2021). Considerations of Covid-19 surveillance for vulnerable populations. <https://www.who.int/publications/i/item/considerations-for-covid-19-surveillance-for-vulnerable-populations>

The World Bank (2021). Covid-19 and Mental Health in Vulnerable Populations: A Narrative Review- Saudi Center for Disease Control and Prevention (English). <https://documents.worldbank.org/en/publication/documents-reports/documentdetail/206591618812759018/covid-19-and-mental-health-in-vulnerable-populations-a-narrative-review-saudi-center-for-disease-control-and-prevention>

8. Clearer justification is needed beyond your apparent decision to do this review because someone else had already reviewed effectiveness.

Response: Thank you for the comment. We acknowledge that there have been published studies examining the effectiveness of social distancing on the general population. However, to the best of our knowledge, no study has systematically reviewed global evidence depicting the various impacts, particularly, the negative effects of social distancing on the vulnerable populations as of July 2022. We see this as an opportunity to fill a major research gap that is important and critical to the policy responses towards the Covid-19 pandemic. Our findings emphasise the need for policymakers and practitioners to pay more attention to addressing the needs and improving the welfare of the vulnerable populations as the world transits into endemicity. Furthermore, the insights and recommendations provided in this research would allow countries to put more efforts and resources into strengthening health provisions and social safety nets for vulnerable populations to better prepare for future public health emergencies.

9. For public health research, put your aim and objectives at the end of your introduction and move your research question/s to Methods. Regarding your questions, why did you only look at negative affects and ringfencing and how are you defining ringfencing?

Response: Thank you for your suggestions. We have incorporated the objectives of this review at the end of the 'Introduction' section, and moved the review questions to the 'Methods' section. As we have mentioned in the response for Q8 above, we focused on the negative effects of physical distancing and ringfencing as these are the research gaps identified. As ringfencing means protecting something by putting limits on it so that it can only be used for a particular purpose or by a particular group. In the context of this study we defined 'ringfencing' as 'a form of sectoral lockdown so that all people within a sector or a location can minimise further interactions with the public' (Tan 2021). For example, restricting external visitors to nursing homes and prisons during Covid-19 are some commonly deployed ringfencing measures during the pandemic. Essentially, ringfencing is a form of physical distancing strategy which aims to limit Covid-19 transmission and contain their spread.

Reference:

Tan, C. (2021). "Ring-fencing intensified to avoid second Covid-19 circuit breaker in Singapore". Accessed from URL: <https://www.straitstimes.com/singapore/ring-fencing-intensified-to-avoid-a-second-circuit-breaker>

10. Conceptualisation: good to include this, but it is more a description of your chosen definition than a theorisation or conceptualisation. So, correct your terminology or revise this and in either case it belongs in the first sub-section of your Methods

Response: Thank you for the comment. We have relocated this section to be the first sub-section of Methods. According to the suggestion, we reconsidered the terminology and removed the term "conceptualisation". The sub-section title became "Types and descriptions of vulnerability" to indicate that it describes the vulnerable populations and vulnerabilities defined in literature.

Methods

11. This actually reads like a hybrid scoping review/systematic review and you would have been better off doing a scoping review. So, overall this section needs a lot of work and should be restructured roughly as follows:

Response: We respectfully disagree with your comment here. We have adhered to the rigorous PRISMA guidelines for a systematic review, which include developing a search string, systematically screening for studies that fulfil the inclusion and exclusion criteria, extracting the data for all included studies, critically appraising all the included studies (incorporating the reviewer's comment in Q.15 below) and applied a framework synthesis approach to analyse our data (Brunton et al. 2020).

Nevertheless, we appreciate the comments and instructions given below (Q.12 to Q.17) and we have made substantial efforts to revise the 'Methods' section accordingly.

Brunton, G., Oliver, S. & Thomas, J. Innovations in framework synthesis as a systematic review method. *Res. Synth. Methods* 11, 316–330 (2020).

12. First sub-section (Study design) should identify the type of review and why chosen, any

theory/conceptualisation, study definitions (e.g. table 1 including all, not just vulnerabilities), and research questions.

Response: Thank you for your comment. We have written the first sub-section (Study design) based on your suggestions, justifying why systematic review was chosen as the method in this review, why do we focus on vulnerable populations and stating the review questions. This subsection reads as follows:-

A systematic review was conducted to examine the types and nature of physical distancing measures implemented across the world, and the extent to which they have negatively impacted vulnerable populations. This approach was chosen as it allows the evidence search, study selection, study appraisal and analysis of data to be done systematically to tease out the collective impacts of physical distancing measures on vulnerable populations and to examine nuances of these impacts in different jurisdictions. A consolidation of global evidence is timely due to the rapid emergence of evidence since the beginning of the pandemic, and the lack of focus on the invisible and hidden impacts of physical distancing interventions – which have been applied nearly universally – on vulnerable populations that are already prone to many disadvantages due to pre-existing socio-economic fault lines. In this systematic review, we posed the following review questions: (i) What physical distancing measures or interventions have been implemented and have they negatively impacted vulnerable populations? (ii) What ringfenced measures have been designed to protect vulnerable populations during the Covid-19 pandemic and how have they been implemented?

13. Second sub-section (e.g. Search strategy/Source identification) should justify the databases searched as well as listing them, and provide an example search syntax as narrative or table (don't send readers to another appendix).

Response: Thank you for your comment. We have rewritten this sub-section based on your suggestion. This sub-section (Search Strategy and Data Sources) reads as follows:-

From March 2021 to April 2021, we searched ten databases to identify articles that could potentially be relevant to the issue of physical distancing measures. The databases were PubMed, Scopus, Web of Science, ProQuest, ProQuest Coronavirus Research Database, Embase, Educational Resource Information Center database, LITCOVID, the Cochrane database of systematic reviews, and WHO's database of global literature on coronavirus disease. We developed the search strategy and search strings (Table 5) with the help of an experienced information specialist. The key terms used for the literature search focused on the themes of Covid-19 and physical distancing policy measures.

Table 5 Key terms included in search strings

Concepts	Key terms in search strings
Covid-19	"ncov" OR "2019 ncov" OR "Covid-19" OR "Covid19" OR "Covid-2019" OR "Covid2019" OR "sars-cov-2" OR "sars cov-2" OR "sarscov2" OR "sarscov-2" OR "sars-coronavirus-2" OR "sars corona virus" OR "sars-like coronavirus" OR "novel coronavirus" OR "novel corona virus" OR "Covid*" OR "coronavirus 2" OR "coronavirus infection*" OR "coronavirus disease" OR "corona virus disease" OR "new coronavirus" OR "new corona virus" OR "new coronaviruses" OR "novel coronaviruses" OR "severe acute respiratory syndrome coronavirus 2" OR "coronavirus" OR "sars-cov"
Social distancing	"social distancing" OR "social isolation" OR "physical distancing" OR "physical distance" OR "safe distancing" OR "lockdown" OR "lock down" OR "quarantine" OR "stay-at-home"

	OR "stay at home" OR "self isolation" OR "self-isolation" OR "remote work" OR "school closure" OR "workplace closure"
Policy measure	"act" OR "design" OR "govern*" OR "intervention" OR "law" OR "legislation" OR "politics" OR "regulation" OR "policy" OR "policies" OR "policy measure" OR "policy instrument" OR "policy mix" OR "policy bundle" OR "policy package"

14. Third sub-section (Screening) should include a table of your eligibility criteria and explain how you conducted double screening of both title/abstracts and full texts. As I don't know your eligibility criteria, I can't judge whether they are reasonable. Additionally, more detail is needed as it's unclear to me how you shared out title/abstract screening, if/how you screened full texts, and how your <10% discrepancy was estimated. This appears to have used a convenience rather than systematic approach.

Response: Thank you for your comment. We have rewritten this sub-section based on your suggestion. This sub-section (Eligibility Criteria and Screening Processes) reads as follows:-

Table 6 Inclusion and exclusion criteria for screening of relevant studies

Inclusion criteria
1) Studies examining vulnerable populations as populations of interest.
2) Studies examining various physical distancing measures from the public policy and/or legal perspectives.
3) Peer-reviewed studies (empirical, conceptual and review studies), policy briefs, reports, editorials, commentaries, perspectives and letters.
4) Studies employing jurisdictions (prefecture/district/country/state/province, single country, multi-countries) as a unit of analysis.
5) Studies employing quantitative, qualitative or mixed-methods research designs.
6) Studies published as full-text articles in the English language between November 2019 and April 2022.
Exclusion criteria
1) Studies examining physical distancing measures but that do not mention its impacts on vulnerable populations.
2) Clinical studies on the Covid-19 pandemic without public policy and/or law dimensions.
3) Studies published before November 2019.
4) Studies for which full-text articles are not accessible or that are not published in the English language.

The screening process involved screening the titles and abstracts to select articles relevant to vulnerable population groups, following the full inclusion and exclusion criteria and referring to the categories of vulnerabilities listed in Table 4. The first author conducted the entire screening process independently to identify relevant studies, while the third author randomly selected more than half of the records to screen independently. Both authors went through two iterations to achieve less than 10% discrepancies. All discrepancies were resolved through a detailed discussion among all three authors. Thereafter, the third author again cross-checked the records chosen by the first author to achieve final agreement on the included studies. Full texts of the identified relevant articles were later retrieved for data extraction.

15. Fourth sub-section (Quality assessment) is missing and should be included.

Response: Thank you for your comment. We have included this sub-section based on your suggestion. This sub-section (Quality assessment) reads as follows:

We applied the Crowe Critical Appraisal Tool (CCAT) to assess the quality of each included study. CCAT is a relevant tool for study appraisal in this review as it offers a high degree of reliability in cases where study designs are highly heterogeneous (i.e. quantitative, qualitative and mixed-methods studies).^{242,243} The CCAT has eight category items (preliminaries, introduction, design, sampling, data collection, ethical matters, results, and discussion), with each category allocated five points, and it has a total aggregate score of 40 (detailed descriptions of how each category item can be scored are available in Supplementary Table 4).

16. Fifth sub-section (Data extraction) should list your extraction headings and justify your approach to dividing the task between authors, as again this does not appear systematic.

Response: Thank you for your comment. We have rewritten this sub-section based on your suggestion. This sub-section (Data Extraction) reads as follows:-

Data extraction followed a predesigned data extraction template created through ongoing discussion among all the authors. To ensure quality control, all three authors piloted the data extraction practices for the first 10% of the identified full-text articles. The authors then discussed the data extraction results to build a consistent understanding of the aim and scope of the review. Thereafter, the first author extracted data from the remaining articles, and the other two authors validated a random selection of the data extraction results to ensure consistency.

17. Sixth sub-section (Analysis) should identify and justify your analysis or synthesis method (e.g. line-by-line reading is not a type of analysis) and clarify who did what.

Response: Thank you for your comment. We have rewritten this sub-section based on your suggestion. This sub-section (Data Analysis) reads as follows:-

The data synthesis involved intensive line-by-line reading of the extracted qualitative information by all the authors. We grouped the studies by categories of vulnerable population groups examined using a framework synthesis approach. Framework synthesis enables structured analysis to be done by following a five-stage approach (familiarisation of the issue, framework selection, indexing, charting and finally mapping and interpretation). It is a versatile analytical approach that accounts for heterogeneity in the types of study included (quantitative, qualitative and mixed method) and is a suitable approach when theory is nascent and emergent which was the case in this review.²⁴⁴ At every stage of the analysis, discussions were held to achieve final agreement on the results.

Findings

18. Such a large number of duplicate and irrelevant sources suggests issues with your database selection and search syntax. However, I can't judge these as they aren't included. Your screening process (which should anyway be in Methods) is unclear. Did you do an initial title-only screening and if so, why? First should be title/abstract, second should be full text screening. Please clarify.

Response: Thank you for the comment. We have conducted an extensive search from ten databases. It is also the reason why we had many duplicates. For instance, due to the nature of the databases, Scopus and PubMed have many overlapping articles. We clarified the screening process within the team. After duplicates were removed we carried out title/abstract screening in the first round which then was followed by full text screening.

19. Fig2 is too difficult to read. Please revise.

Response: Thank you for the comment. We have updated the data in Figure 2 and changed the layout and resolution to enhance the visibility.

20. Findings section needs some clarification. Please restructure sub-sections in alignment with your synthesis/analysis (which you will have explained in Methods). Ideally, give an indication in each sub-section of how many sources discussed each topic and how social groups were defined. Your findings are entirely descriptive - which isn't problematic in itself - but it's therefore even more difficult to determine how/if you prioritised and structured your findings - this is unfortunate, as the information you provide is interesting.

Response: Thank you for this comment. Following review 1 and 2's suggestions, we have revised the entire Findings Section to enhance its clarity. We have defined different vulnerable populations and provided more examples to contextualise the negative effects of social distancing faced by each vulnerable population. Our targeted systematic update of the literature, conducted following the suggestion by reviewer 1, also enhances the nuances of the findings, and helps to expand the examples described in some of the vulnerable population groups, albeit this endeavour does not change our results fundamentally.

According to second reviewer's suggestions, we have added some nuances on how the social groups were defined (see Supplementary Text for more details on the definition of the social groups). For instance:

Migrant workers are defined as "a person who is to be engaged, is engaged or has been engaged in a remunerated activity in a state of which he or she is not a national" ²⁹⁰. They move away from their place of usual residence across the international border to a different country or simply to a different place within a country ²⁹¹.

A disability is defined as any physical or mental condition (impairment) that results in a limitation on a person's activity (difficulty in undertaking certain activities, e.g., difficulty seeing, hearing, walking, or problem-solving) or a restriction on their participation (difficulty participating in normal daily activities, e.g., working, social activities, or accessing healthcare and preventive services).¹⁴²

Based on reviewer 2's suggestion, we have added some statistics to the main manuscript, such as how many studies discussed each topic.

The majority of studies in our dataset focus on children/students and low-income people (96 studies and 58 studies, respectively), following by studies on the elderly (n=37), domestic violence victims (n=16), people with disabilities (n=15), migrant workers (n=14), refugees (n=14), LGBTQ+ people (n=11), ethnic minorities (n=10), sex workers (n=9), and prisoners (n=7). There are five studies on vulnerable groups in general. The classification we apply is not exclusive, and vulnerabilities can be

intertwined, such as children with disabilities or children who are vulnerable to domestic violence.

See Supplementary Table 5~ Table 6 for details on the distribution of included studies across each topic and each country.

Supplementary Table 5. Distribution of studies across countries and vulnerable populations

Vulnerable population	Number of studies	Distribution across countries	Vulnerable population	Number of studies	Distribution across countries
Elderly	37	Not specific (1 study):²¹ Global (1 study):²² EU countries (1 study):²³ Europe (4 studies):^{24,25,26,27} France (1 study):²⁸ Italy (3 studies):^{29,30,31} Spain (4 studies):^{32,33,34,35} Germany (1 study):⁶ Turkey (1 study):³⁶ Finland (1 study):³⁷ Sweden (1 study):³⁸ South Korea (1 study):³⁹ Brazil (1 study):⁴⁰ China (2 studies):^{41,42} India (3 studies):⁴³⁻⁴⁵ US (4 studies):^{46-48,49} UK (3 studies):^{50,51,52} Canada (1 study):⁵³ Centro American countries (1 study):⁵⁴ sub-Saharan Africa (1 study):⁵⁵ Australia (1 study):⁵⁶	Children/students	96	Not specific (7 studies):⁵⁷⁻⁶³ Global (9 studies):^{64-68, 69,70,71} 9 countries in Europe (1 study):⁷² Low- and lower-middle-income countries (1 study):⁷³ Germany, Austria, Switzerland (1 study):⁷⁴ Germany (1 study):⁷⁵ UK (11 studies):^{76-82,83,84,85,86} US (19 studies):^{87-104,105} Brazil (2 studies):^{106,107} Finland and Sweden (1 study):¹⁰⁸ Italy (3 studies):^{109,110,111} Norway (2 studies):^{112,113-114} Denmark (1 study):¹¹⁵ Belgium (1 study):¹¹⁶ Spain (1 study):¹¹⁷ Turkey (1 study):¹¹⁸ Portugal (1 study):¹¹⁹ Egypt (1 study):¹²⁰ France (2 studies):^{121,122} Netherlands (2 studies):^{123,124} Australia (3 studies):^{125,126,127} Canada (1 study):¹²⁸ China (1 study):¹²⁹ India (11 studies):^{130-135,136,137,138,139,140} Bangladesh (1 study):¹⁴¹ Japan (3 studies):¹⁴²⁻¹⁴⁴ Nepal (1 study):¹⁴⁵ Pakistan (1 study):¹⁴⁶ South Africa (3 studies):¹⁴⁷⁻¹⁴⁹ Namibia (1 study):¹⁵⁰ Kenya (1 study):¹⁵¹ Jordan (1 study):¹⁵²
Low-income people	58	Not specific (3 studies):¹⁵³⁻¹⁵⁵ Global (2 studies):^{64,156} UK (4 studies):^{76,157,158,159} US (11 studies):¹⁶⁰⁻¹⁷⁰ India (11 studies):^{2,45,135,171-175,176,177,178} Chile (1 study):¹⁷⁹ Bangladesh (1 study):¹⁸⁰ Brazil (1 study):¹⁸¹ Canada (2 studies):^{128,182} China (2 studies):^{183,184} Philippines (1 study):¹⁸⁵ Italy (1 study):¹⁸⁶ Mexico (1 study):¹⁸⁷ Nigeria, Rwanda, South Africa, and Uganda (1 study):¹⁸⁸ Kenya (1 study):¹⁸⁹ Namibia (1 study):¹⁵⁰ Kenya, South Africa (1 study):¹⁹⁰ Africa (1 study):¹⁹¹ Kenya, South Africa, Nigeria (1 study):¹⁹² South Africa (6 studies):^{148,149,193-196} Nigeria (2 studies):^{197,198} Uganda (1 study):¹⁹⁹ Zimbabwe (1 study):²⁰⁰ New Zealand (1 study):²⁰¹	Migrant workers	14	Not specific (1 study):¹⁵⁴ India (5 studies):^{44,135,202,203,204} South Africa (1 study):¹⁴⁷ Malta (1 study):²⁰⁵ Singapore (2 studies):^{206,207} Bangladesh (1 study):²⁰⁸ France (1 study):²⁰⁹ UK (1 study):²¹⁰ Mexico (1 study):²¹¹

Vulnerable population	Number of studies	Distribution across countries	Vulnerable population	Number of studies	Distribution across countries
Prisoners	7	Global (1 study): ²¹² Africa (1 study): ¹⁹¹ Portugal (1 study): ²¹³ Sweden (1 study): ²¹⁴ UK (1 study): ²¹⁵ US (2 study): ^{216, 217}	People with disabilities	15	Global (1 study): ¹⁶⁶ Australia (1 study): ¹²⁶ India (1 study): ¹⁷⁴ Italy (4 studies): ^{109,218,219,218} Spain (1 study): ²²⁰ US (3 studies): ^{104,221,222} Belgium (1 study): ¹¹⁶ UK (1 study): ²²³ Greece (1 study): ²²⁴ Romania (1 study): ²²⁵
Sex workers	9	Not specific (1 study): ²²⁶ Africa (1 study): ²²⁷ India (2 studies): ^{228,229} Kenya (2 studies): ^{230,231} Thailand (1 study): ²³² Singapore (1 study): ²³³ Nigeria (1 study): ²³⁴	Domestic violence	16	Not specific (1 study): ²³⁵ Global (1 study): ²³⁶ Bangladesh (3 studies): ^{237,238,239} India (2 studies): ^{45, 240} UK (2 studies): ^{241,242} Spain (1 study): ²⁴³ US (2 studies): ^{93,244} South Africa (1 study): ²⁴⁵ Guatemala (1 study): ²⁴⁶ Democratic Republic of Congo (1 study): ²⁴⁷ Norway (1 study): ²⁴⁸
Refugees	14	Not specific (4 studies): ²⁴⁹⁻²⁵² Africa (1 study): ¹⁹¹ Germany (1 study): ²⁵³ Greece (2 studies): ^{254,255} South Africa (1 study): ²⁵⁶ UK (1 study): ²⁵⁷ US (2 studies): ^{258,259} Jordan (1 study): ²⁶⁰ Italy (1 study): ²⁶¹	Ethnic minority	10	India (1 study): ¹⁷⁵ UK (5 studies): ^{4,157,262,263,264} US (4 studies): ^{89,166,265,266}
LGBTQ+	11	Not specific (2 studies): ^{267,268} India (1 study): ²⁶⁹ UK (2 studies): ^{270,52} US (5 studies): ²⁷¹⁻²⁷⁵ Brazil (1 study): ²⁷⁶	General vulnerable	5	Studies discussing more than three vulnerable groups or talking about vulnerable populations in general without specifying detailed population groups. UK (1 study): ²⁷⁷ South Korea (1 study): ²⁷⁸ Austria (2 studies): ^{279,280} Netherlands (1 study): ²⁸¹
Total				265	

Note: 1) "Not specific" indicates studies not mentioning specific geographic study scopes.

Supplementary Table 6. Summarized characteristics of included studies

Indicator	Details	Number of studies	Percent (% out of 179 studies)
Country of origin	Single country	218	82.26
	• US	48	18.11
	• India	30	11.32
	• UK	27	10.19
	Multiple countries/Global	28	10.57
	Not specific	19	7.17
Population of interest	Elderly	37	13.96
	Children/students	96	36.23
	Low-income people	58	21.89
	Migrant workers	14	5.28
	Prisoners	7	2.64
	People with disability	15	5.66
	Sex workers	9	3.40
	Domestic violence	16	6.04
	Refugees	14	5.28
	Ethnic minority	10	3.77
	LGBTQ+	11	4.15

Discussion

21. Your paragraph on your study's contributions needs more detail, e.g. it should include whether anything in your findings is new or if it reinforces existing knowledge.

Response: Thank you for this comment. Following your point we further elaborated on the contributions. There have been review studies on social distancing measures and their effectiveness, but few reviews have focused specifically on vulnerable populations. The few studies which have examined vulnerable populations have mainly focused on distinctive vulnerable groups. This systematic review, to the best of our knowledge, is one of the earliest reviews that attempts to broaden the scope of vulnerable populations by studying the unintended consequences of physical distancing policies and practices on 11 vulnerable groups of social distancing policies and practices.

22. Don't include new data or data tables in your Discussion - these should be in Findings

Response: Thank you for the suggestion. We have shifted this table to the end of Finding Section, under a new sub-section titled "Summary of findings".

23. Include a limitations sub-section

Response: Thank you for your suggestion. We have included some discussion on the limitations of this review in the 'Discussion' section. It reads as follows:

This systematic review has two potential limitations. First, we may have missed out a small subset of studies during the systematic evidence search process due to the various ways vulnerabilities and vulnerable populations are defined and conceptualised. Second, we did not attempt to tease out the jurisdictional differences pertaining to the negative effects experienced by the 11 vulnerable populations examined in this review. This presents an opportunity for future research.

24. Include a sub-section on how study findings relate to existing knowledge and general relevance for policy, practice and further research as otherwise your recommendations table comes out of nowhere. Additionally, Table 4 seems more like a list of preferences/wishes rather than being prioritised by feasibility or value or grounded in evidence.

Level of interest: An article of relevance to those with similar interests

Quality of written English: Good

Statistical review: No

Response: Thank you for your comment. We added a sub-section 'Broader Impact' in Discussion section about how study findings relate to existing knowledge and general relevance for policy.

Table 4 lists the best practices summarised from the included studies. To enhance clarity, we added citations to each policy recommendation in Table 4 to show the studies they refer to. For instance, policy recommendation 'A comprehensive support package for the vulnerable populations to ensure reliable access to necessities, such as food, housing/shelter, electricity, water and sanitation.^{85,93,104,106,157,162,222}' is based on several studies.

85. Barnett-Howell, Z., Mobarak, A. M. & Watson, O. J. The benefits and costs of social distancing in high- and low-income countries. *Trans R Soc Trop Med Hyg* 1–13 (2021).
93. Singh, I., Singh, J. & Baruah, A. Income and employment changes under COVID-19 lockdown: A study of Urban Punjab. *Millenn. Asia* 11, 391–412 (2020).
104. Iwuoha, V. C. & Aniche, E. T. Covid-19 lockdown and physical distancing policies are elitist: towards an indigenous (Afro-centred) approach to containing the pandemic in sub-urban slums in Nigeria. *Local Environ.* **25**, 631–640 (2020).
106. Sahu, M. & Dobe, M. How the largest slum in india flattened the covid curve? A case study. *South East. Eur. J. Public Heal.* **14**, 1–15 (2020).
157. Janyam, S. *et al.* Protecting sex workers in Thailand during the COVID-19 pandemic: opportunities to build back better. *WHO South-East Asia J. public Heal.* **9**, 100–103 (2020).
162. Callander, D. *et al.* Sex workers are returning to work and require enhanced support in the face of COVID-19: Results from a longitudinal analysis of online sex work activity and a content analysis of safer sex work guidelines. *Sex. Health* **17**, 384–386 (2020).
222. Mukumbang, F. C., Ambe, A. N. & Adebisi, B. O. Unspoken inequality: How COVID-19 has exacerbated existing vulnerabilities of asylum-seekers, refugees, and undocumented migrants in South Africa. *Int. J. Equity Health* **19**, 1–7 (2020).

REVIEWER COMMENTS

Reviewer #1 (Remarks to the Author):

The authors have addressed all of my concerns. Their letter and revisions are responsive in great detail to all of my concerns as well as Reviewer 2's concerns. I was particularly positive about the addition of new articles to cover the period through June 2022.

Reviewer #3 (Remarks to the Author):

Thank you for the opportunity to review the authors' responses to the original Reviewer 2's comments.

I believe the authors did a diligent and rigorous job in responding to the comments, particularly on Reviewer 2's request to clarify the 'social distancing measures' and the 'vulnerable groups'. The authors also introduced valid changes in response to Rev2's request to follow more closely the systematic review's methodology, and added more details on the inclusion and exclusion criteria, the quality assessment, and the search strategy.

In my view, the way it is, this manuscript is a valid and interesting review of the evidence around the negative effects of covid-related physical distancing measures, but I must agree with Rev2's original assessment that this is more a scoping review, and it cannot be called a systematic review as it does not follow the stated standards.

First of all, protocols for systematic reviews are typically registered and cleared in the PROSPERO platform, but I did not see any mention of that in the manuscript.

Secondly, systematic reviews typically follow a PICO approach, where the evidence on the effect of a specific intervention is associated with a specific outcome. But in this review, vague physical distancing interventions are somewhat related to similarly vague negative effects (although the new Table 2 helps clarifying such measures). As a result, one is left wondering what specific policy measure has brought what health outcome, and in what specific country context.

I share Reviewer 1's praise for such a refreshing (ie, non meta-analytical), gigantic, and open study; but in my opinion, this should have been designed in the first place as a scoping review (as correctly pointed out by Reviewer 2), which can be as rigorous as a systematic one, while leaving enough space to scope the lay of the land, and take stock of what is known on a specific, broad topic. Following one of Reviewer 1's remarks about the study only covering publications until 2021, I would also argue that this should have been designed as a 'Living scoping review', left open for regular updates, and leading possibly to a systematic review of specific interventions on specific health outcomes.

Response to Referees letter NCOMMS-22-08041B

Revision 2 – A scoping review of the impacts of physical distancing measures on vulnerable population groups

Responses to the reviewers,

We would like to thank the reviewers for their time, helpful comments and suggestions.

We were pleased that we have been given the opportunity to further refine the manuscript, and that, in the words of reviewer 1 “The authors have addressed all of my concerns. Their letter and revisions are responsive in great detail to all of my concerns as well as Reviewer 2's concerns.”, and in words of reviewer 3 “the authors did a diligent and rigorous job in responding to the comments, particularly on Reviewer 2 request to clarify the ‘social distancing measures’ and the ‘vulnerable groups’” and that “In my view, the way it is, this manuscript is a valid and interesting review of the evidence around the negative effects of covid-related physical distancing measures”.

All comments are appreciated and below we describe how we accounted for each of the comments and, overall, how we improved the quality of the manuscript and clarified its limitations. Reviewer 3 is in agreement with the previous comments of Reviewer 2 that the paper should be reoriented as a scoping review rather than a systematic review. Based on the comment we have made extensive changes that are highlighted using tracked changes in the revised manuscript and/or below.

The Authors

REVIEWER COMMENTS

Reviewer #1

The authors have addressed all of my concerns. Their letter and revisions are responsive in great detail to all of my concerns as well as Reviewer 2's concerns. I was particularly positive about the addition of new articles to cover the period through June 2022.

Response: We thank Reviewer #1 for the affirmation of our extensive revisions that addressed both reviewers' concerns in the previous round of revision, as well as the positive comment on the addition of new articles to our review to cover the period through June 2022.

Reviewer #3

Thank you for the opportunity to review the authors' responses to the original Reviewer 2's comments.

I believe the authors did a diligent and rigorous job in responding to the comments, particularly on Reviewer 2's request to clarify the 'social distancing measures' and the 'vulnerable groups'. The authors also introduced valid changes in response to Rev2's request to follow more closely the systematic review's methodology, and added more details on the inclusion and exclusion criteria, the quality assessment, and the search strategy.

Response: Thank you very much for this comment and your recognition of our extensive efforts to address comments of Reviewer 2 in clarifying social distancing measures and vulnerable groups as well as improving the details of the methodology.

In my view, the way it is, this manuscript is a valid and interesting review of the evidence around the negative effects of covid-related physical distancing measures, but I must agree with Rev2's original assessment that this is more a scoping review, and it cannot be called a systematic review as it does not follow the stated standards.

First of all, protocols for systematic reviews are typically registered and cleared in the PROSPERO platform, but I did not see any mention of that in the manuscript.

Secondly, systematic reviews typically follow a PICO approach, where the evidence on the effect of a specific intervention is associated with a specific outcome. But in this review, vague physical distancing interventions are somewhat related to similarly vague negative effects (although the new Table 2 helps clarifying such measures). As a result, one is left wondering what specific policy measure has brought what health outcome, and in what specific country context.

I share Reviewer 1's praise for such a refreshing (ie, non meta-analytical), gigantic, and open study; but in my opinion, this should have been designed in the first place as a scoping review (as correctly pointed out by Reviewer 2), which can be as rigorous as a systematic one, while leaving enough space to scope the lay of the land, and take stock of what is known on a specific, broad topic. Following one of Reviewer 1's remarks about the study only covering publications until 2021, I would also argue that this should have been designed as a 'Living scoping review', left open for regular updates, and leading possibly to a systematic review of specific interventions on specific health outcomes.

Response: Thank you very much and we appreciate the comment by Reviewer #3 regarding systematic vs scoping review.

To address this comment as well as the comment raised by the Reviewer 2 and the editor earlier, we changed the methodological framing of this paper as a scoping review. We believe this is a better reflection of the extent and scope of work that we have done and as well as aligning closer to our findings. Consequent to this decision, we have removed two sub-sections in the paper ('Quality Assessment' under the methods and 'Critical Appraisal' under the results section) to better align with the requirement of a scoping review.